# Wind sensing with drone mounted wind lidars: proof of concept

Nikola Vasiljević[1], Michael Harris[2], Anders Tegtmeier Pedersen[1], Gunhild Rolighed Thorsen[1], Mark Pitter[2], Jane Harris[3], Kieran Bajpai[3], and Michael Courtney[1]

[1]Technical University of Denmark - DTU Wind Energy, Frederiksborgvej 399, Building 118-VEA, 4000 Roskilde, Denmark
[2]ZX Lidars, The Old Barns, Fairoaks Farm, Hollybush, Ledbury HR8 1EU U
[3]University of Exeter, Department of Physics, UK

**Correspondence:** Nikola Vasiljević (niva@dtu.dk)

**Abstract.** The fusion of drone and wind lidar technology introduces the exciting possibility of performing high-quality wind measurements virtually anywhere. We present a proof of concept (POC) drone-lidar system and report results from several test campaigns that demonstrate its ability to measure accurate wind speeds. The POC system is based on a dual-telescope Continuous Wave (CW) lidar, with drone-borne telescopes and ground-based opto-electronics. Commercially available drone and gimbal units are employed. The demonstration campaigns started with a series of comparisons of the wind speed measurements acquired by the POC system to simultaneous measurements performed by nearby mast based sensors. On average, an agreement to about 0.1 m/s between mast- and drone- based measurements of the horizontal wind speed is found. Subsequently the extent of the flow disturbance caused by the drone downwash was investigated. These tests vindicated the somewhat conservative choice of lidar measurement range made for the initial wind speed comparisons. Overall, the excellent results obtained without any drone motion correction and with fairly primitive drone position control indicate the potential of drone-lidar systems in terms of accuracy and applications. The next steps in the development are outlined and several potential applications are discussed.

## 1 Introduction

For many years, wind energy has been one of the fastest growing power production technologies in Europe. Based on the average predictions the wind will deliver a quarter of power demands in Europe by 2030 (EWEA, 2015). The annual installed capacity of wind energy has seen a consistent growth of about 5%. Wind power accounted for 55.2 % of total installed power in Europe in 2017 (WindEurope, 2018), and overall represents 18% of total installed power generation capacity. Wind energy satisfies about 11.6% of EU total electricity demands (WindEurope, 2018).

The growth of wind energy is heavily dependent on accurate wind speed measurements, which are essential for various applications such as the prediction of annual energy production (AEP) for wind farms and power curve verifications. Given

the size of modern wind turbines, which today operate between 60 to 220 m above ground level (agl), and the need to capture wind across the entire rotor plane, there is a demand for measurements at heights well above 100 m.

However, already beyond 60 m agl conventional in-situ wind speed measurements require costly towers. Offshore especially, any tower-based measurement is usually prohibitively expensive. These economic realities have fostered the development of ground-based (Courtney et al., 2008), nacelle-mounted (Borraccino et al., 2016), floating (Gottschall et al., 2017) and multi-lidar (Vasiljevic et al., 2016) systems all of which are now competitive in various wind energy use cases.

Wind lidars, unlike conventional anemometry, retrieve information about the wind without being in direct contact with the moving air. This is done by remotely probing the atmosphere using laser light. Two types of Doppler wind lidars are available distinguished by how they probe the atmosphere (Courtney et al., 2008). CW lidars stream a continuous laser beam, focused at the location of interest. Pulsed lidars stream a burst of short laser pulses and can retrieve wind speeds at various distances along this line. CW lidars have a typical range from 10 to 250 m with a high measurement frequency (several hundred scans per second are possible in good conditions) at a single range. Pulsed lidars have a range from 50 m up to 10 km with a low measurement frequency (around 1 Hz) but the ability to measure at many ranges simultaneously.

Since the introduction of wind lidars in wind energy domain in 2003 (Harris et al., 2007) they have been extensively used in industry and research for resource assessment Krishnamurthy et al. (2013), wind turbine power curve measurements Wagner et al. (2014), feed-forward control Simley et al. (2018), wake measurements Herges et al. (2017), wake steering Fleming et al. (2017) and short-term forecast Würth et al. (2019). Over the last decade wind lidars have became an established measurement technique within the wind energy community. Beyond the wind energy domain, wind lidars have been used in wind engineering especially to provide wind measurements to properly estimate wind loading on bridges (Cheynet et al., 2017). Historically, (since the 1980s) wind lidars have been used extensively in variety of atmospheric science studies (e.g., McCarthy et al., 1982; Newsom et al., 2008; Collier et al., 2005; Grubišić et al., 2008; Fernando et al., 2015).

Even though wind lidars are cost-attractive instruments for measurements beyond 100-m they are still relatively expensive. A minimum cost of an accurate wind lidar is about 60 k€ and 1-2 M€ for on-shore and offshore applications respectively.

Also, there are circumstances where wind lidars can experience difficulties, in which traditional in-situ measurements can measure successfully. Lidar range is influenced by the atmospheric conditions (i.e., aerosols concentration), which impacts the data availability (in certain locations lidar data availability can fall below mast-based sensor data availability). For example, clouds, fog and snow are highly attenuating for the laser beam which limits the lidar range, and thus data availability. Furthermore, any precipitation will affect the wind speed measurements by lidars. Specifically the vertical component of the wind will be biased since the lidar will dominantly measure the fall velocity of the precipitation (Lindelöw, 2009).

Beyond the range of 50 m, independent of the lidar type, the effects of measurement volume become significant, and this limits the lidar applicability for the assessment of small-scale turbulence. Single lidars only measure the projection of the wind velocity on the laser propagation path (i.e., radial velocity), which in fact requires assumption of the horizontal homogeneity of the flow to reconstruct the wind speed (Browning and Wexler, 1968; Strauch et al., 1987). This bounds the usability of single lidars to offshore (Peña et al., 2008) and sites with simple topography on-shore (Courtney et al., 2008). Therefore, when the flow is complex, as it is the case in more than 50% of the onshore sites (e.g., hilly terrain) with good wind resources (Bingöl,

2010), multi-lidar instruments such as long-range (Vasiljevic et al., 2016) or short-range (Sjöholm et al., 2014) WindScanner systems are needed to accurately retrieve the full wind flow. This of course drastically increases costs (several lidars) as well the complexity of measurements (installation, configuration, synchronisation and monitoring) and corresponding data analysis (processing and integrating several datasets). This is one of the main reasons why multi-lidar measurements are mainly used in research projects for short-term measurement campaigns (e.g., Lundquist et al., 2017; Mann et al., 2017; Fernando et al., 2019).

To formulate the research problem, currently there are no measurement solutions that would provide:

– low cost yet accurate measurements in all sites (offshore, flat and complex terrain) and at altitudes where modern wind turbines operate (60 to 220 m agl),

– simple approach to perform measurements in different locations in the atmosphere (mobile measurements)

– high frequency measurements with a small probe volume (i.e., simultaneous mean flow and turbulence measurements), and

– high availability of data (i.e., not to be hindered by fog, low clouds, etc.).

– long measurement duration (e.g., months)

A potential solution for the above formulated problem is to use a Small Unmanned Aircraft System (SUAS), such as multi-copter drones, as a platform for a wind lidar even though currently SUAS cannot offer a long uninterrupted operation. Typically SUAS acquire wind speed information either by utilizing flow sensors such multi-hole pitot tube probes (e.g., Wildmann et al., 2014) or sonic anemometers (e.g., Nolan et al., 2018) or without flow sensors by measurements and conversion of aircraft dynamics (e.g. Neumann and Bartholmai, 2015). For example, studies Neumann and Bartholmai (2015), Palomaki et al. (2017) and Brosy et al. (2017) utilized real-time measurements of multi-copter dynamics to estimate wind speed. These studies reported a good agreement of the estimated wind speed with the speed measured by mast-mounted sonic anemometers, where the sonic and estimated wind speed agreed to about 0.5 to 0.7 m/s for 10 s averaged data (Palomaki et al., 2017; Brosy et al., 2017) and 0.3 m/s for 20 s averaged data (Neumann and Bartholmai, 2015). Brosy et al. (2017) stated that the wind speed estimated using only drone dynamics should not be used as information about atmospheric turbulence since due to the volume drones don't react to the small eddies, and thus this approach cannot capture a full range of wind speeds. In LAPSE-RATE experiment (Barbieri et al., 2019) several multi-copter drones were equipped with sonic anemometers. The calibration flights of such drones (which entail hovering the equipped drones close to masts equipped with sonic anemometers) showed agreement of the mast and drone-based 15-s averaged wind speed measurements to about 0.75 m/s (Nolan et al., 2018). Like in case of multi-copter drones, the aircraft dynamics of fixed-wing SUAS can be used to determine wind speed (Rautenberg et al., 2018) with the wind speed accuracy generally worse than multi-copter drones (Barbieri et al., 2019). As stated in Rautenberg et al. (2018) utilizing a flow sensor such as pitot tube on-board of fixed-wing SUAS generally provides better results. Nevertheless, as reported in Barbieri et al. (2019) an average accuracy of SUAS of both fixed-wing and multi-copter concepts with or without flow sensors is about 0.5 m/s, which for the majority of wind energy applications is not sufficient.

However, if we use a wind lidar as a flow sensor, potentially the accuracy could be improved and also given the ability to acquire turbulence measurements. Equiping SUAS with wind lidars has been suggested in an early study of using SUAS for wind energy applications (Giebel et al., 2012). At the time of the study, it was technically unfeasible to pursue this idea. Roughly a decade later, both the lidar and drone technology have advanced significantly unlocking the potential to explore the proposed idea.

In the concept we propose in this paper the drone would be used to position the lidar in the vicinity of the measurement points and to steer the outgoing laser beam. This would have several radical implications on the wind lidar development.

First, the required maximum range would be in the order of a few meters (i.e., just enough to avoid the impact of the drone downwash on intended measurements of the free flow). Second, since the drone can be used to steer the outgoing laser beam the lidar would not need to have a variable focus or a scanning mechanism. Third, since the drone alone can be used to sense the wind (see Brosy et al. (2017)) we can eliminate an acousto-optic modulator (AOM) from the lidar design because the Doppler shift sign can be calculated from the drone dynamics.

The combination of the above-described implications leads to a significant reduction in the lidar complexity (fewer and cheaper components), size, weight and power consumption, and thus potentially in the overall costs.

The requirements for the drone-mounted lidar can be met by a low-power small-optics CW lidar with a manual focus adjustment. The use of CW technology will allow for a high measurement frequency ($\sim$50 Hz). Due to the expected short focus range the resulting probe length would be rather small ($\sim$10 cm), allowing accurate measurements of both the mean wind and turbulence. Additionally, short-range measurements would not be hindered by fog or clouds. In fact these atmospheric conditions would be favorable due to the substantially increased backscatter, and thus an improvement in signal-to-noise (SNR) ratio and data availability. Additionally, if we have a fully non-tethered drone measurements can be made in difficult locations such as above thick forests that do not have suitable clearings for ground-based lidars.

In this paper we present the preliminary results of realizing the aforementioned drone-lidar system in practice. Specifically we will present the results of the proof of concept (POC) stage of a drone-based wind lidar system development. As such the POC system is developed only to demonstrate the feasibility of the proposed concept, thus to show that the concept has practical potentials.

The paper is organized as follows. Section 2 presents the POC drone-based wind lidar system. Section 3 presents results of several demonstration experiments conducted with the described measurement system. Section 4 discusses the results of the POC stage as well as outlines our future work, while Section 5 provides our concluding remarks.

## 2 Measurement system description

### 2.1 Overview

For the POC, we used a non-production dual-telescope CW lidar system built by ZX Lidars (ZXT2 lidar), and off-the-shelf drone and gimbal system (DJI Matrice 600 Pro and Ronin-MX). The selected drone and gimbal system are typically used in

**Table 1.** Basic specifications of ZXT2 lidar

| | |
|---|---|
| Base unit dimensions | 3U 19" rack unit |
| Length of fibres connecting telescopes to base unit | 100 m |
| Type of fibres | single-mode |
| Telescope dimensions | Length 25 cm by diameter 3 cm |
| Telescope weight | 100 grams |
| Focal range | 0.5m to 50 m |
| Transmitted laser power | 300mW CW per channel |
| Laser wavelength | 1.56 $\mu$m |
| Probe length (FWHM) | Approx 5cm at 3m range |
| Line-of-sight speed range | $\pm$ 0.3 – 39.0 m/s (sign not distinguishable) |
| ADC sample rate | 100 MHz on both channels simultaneously |
| Data rate | 48.83 Hz independent line-of-sight speed values on both channels |
| Number of averages per measurement | 4000 |
| Bin width | 0.1523 m/s |

the motion picture industry, while the lidar was optimized for wind tunnel measurements or for turbine blade mounting, thus the transceiver units (telescopes) are separated from the rest of the lidar.

Instead of mounting the entire lidar to the drone, for the POC we mounted only the telescopes to the gimbal (attached to the drone). The telescopes were connected to the lidar located on the ground using 100-m long optical fibres. The drone was battery powered.

The main reason why the POC system was built as described above was that it did not required any costly development since many of the parts were already built, or readily available off the shelf. Moreover, since we intended to investigate the overall feasibility of the proposed concept, this type of study (i.e., proof-of-concept) is often undertaken on a much lower budget and before investing in the build of a full prototype or product development.

In the text that follows, we will describe each part of the POC measurement system in more details.

## 2.2 Lidar system

The ZXT2 lidar is an experimental system built in 2014 (Neininger, 2017). It has since been used in a variety of trials including several wind tunnel tests and lidar calibration exercises. ZXT2 is a two-channel CW wind lidar system consisting of a 3U 19" rack unit with two separate staring transceiver units (telescopes), allowing simultaneous, continuous and independent line-of-sight (LOS) wind measurements. It achieves this by sharing the laser power 50/50 between the two channels. As a result, the unit exhibits reduced sensitivity compared to a unit that uses 100% of the power in a single channel. The main specifications are listed in Table 1.

The rack unit contains an internal Windows PC; normal operation involves accessing the internal PC via remote desktop. In the drone experiment, the telescopes with 1" aperture were connected to the rack unit with lightweight 100m fibre cables. Four fibres were required to provide the transmit and receive paths on the two channels.

The beam from each telescope is brought to a near-diffraction-limited focus, at a distance that can be manually adjusted on the ground then locked in position. The total laser output can range from 0.3 W to 1.3 W, but here it was set to 0.8 W, giving an output of order 0.3 W on each channel (after accounting for losses). This output level provided excellent signal strength under the conditions experienced during the tests. Care was taken to avoid fibre bend losses during flight operations.

The detector output is sampled at 100 MHz, and Doppler signals are obtained as an average of 4000 independent 512-point FFT spectra. Each averaged spectrum consists of 256 bins, spanning a speed range 0-39 m/s, and allowing calculation of LOS wind speed via different estimation algorithms. For this experiment, a simple median method was used (e.g., Held and Mann, 2018), so that the basic output consists of two simultaneous channels of speed values at nearly 50Hz data rate. ZXT2 is a homodyne system (no AOM), and therefore the sign of the Doppler shift was not detected. For systems that use AOMs, the outgoing laser light frequency is shifted in order to be able to detect the sign of the Doppler shift of the backscattered light. One approach to eliminate an AOM from the lidar design is to have an external source of information about the wind direction, which in case of for example Zephir profiling lidars is accomplished by having an additional wind sensor attached to the lidar (Courtney et al., 2008). All raw spectral data are stored to allow more detailed analysis to be carried out if required.

## 2.3 Drone and gimbal

DJI Matrice 600 Pro is an aerial platform build for various industrial and scientific applications (DJI, 2018). This hexacopter has a solid loading capability and hovering time of about 20 min with a payload of about 5 kg (see Table 2 for the basic drone specifications). The flight time can be significantly improved by converting the drone to be tethered (e.g., power line auto-track tension system). The drone originally comes equipped with three Global Navigation Satellites Systems (GNSS) receivers for position measurements (accuracy of a few meters). The drone has a modular design which allows the integration of both the DJI branded sensors as well as third-party sensors. For boosting the accuracy of position measurements and providing improved measurements of the drone orientation we have additionally equipped the drone with two receivers for real-time differential GNSS measurements (DGNSS) which communicate with a mobile ground based real-time kinematics (RTK) station (i.e., rover). The baseline distance between the two DGNSS antennas of 25 cm allows the orientation measurements accuracy of about $0.8°$ as stated by DJI in the product sheet (DJI, 2017). Beside the GNSS receivers, the drone is equipped with three inertia measurement units (IMU). The combination of RTK and IMU should in principle provide a solid foundation for accurate measurements of the drone 6 degrees of freedom (6DOF) in real-time (i.e., position, orientation, leveling, velocity and acceleration). The drone has been equipped with an on-board camera to provide visual information. DJI provides a well-document software development kit (SDK) and an application programming interface (API) for the drone which allows the development of automated drone applications.

To improve the stability (e.g., damp vibrations), orientation and tilt of the drone-mounted telescopes we have equipped the drone with a three-axis programmable gimbal system (Ronin-MX, see basic specifications in Table 3 extracted from the

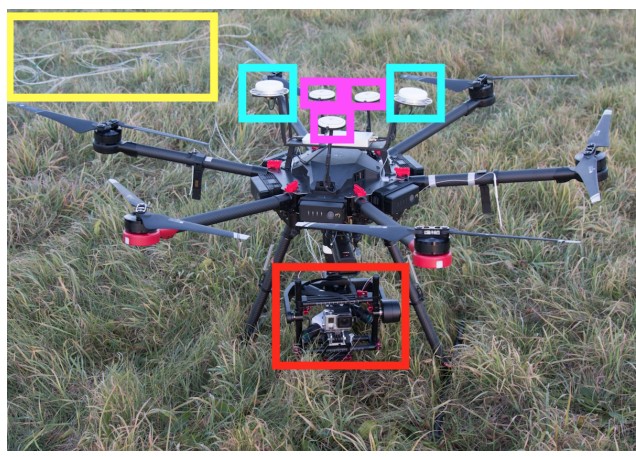

**Figure 1.** Complete system on ground: purple squares - three central GNSS antennas for positioning, turquoise squares - two DGNNS receivers with RTK for real time position correction, yellow square - four joined optical fibres and red square - gimbal with telescopes and GoPRO camera. Picture taken from the photo gallery https://work.courtney.dk/#collection/7.

**Table 2.** Basic specifications of drone from DJI product sheet (DJI, 2018, 2017)

| | |
|---|---|
| Dimensions | 1668 mm (width) × 1518 mm (length)× 727 mm (height) |
| Dimensions (folded) | 437 mm (width)× 402 mm (length)× 553 mm (height) |
| Weight | $\sim$ 10 kg |
| Maximum takeoff weight | $\sim$ 15 kg |
| Hovering time no payload | $\sim$ 35 min |
| Hovering time with $\sim$ 5kg payload | $\sim$ 20 min |
| Operating temperature | -10$^\circ$C to 40$^\circ$C |
| Maximum angular velocity | Tilt: 300$^\circ$/s, Yaw: 150$^\circ$/s |
| Maximum ascent velocity | 5 m/s |
| Maximum descent velocity | 3 m/s |
| Maximum flying velocity in zero wind | 18 m/s |
| Maximum wind resistance (recommended) | 8 m/s |
| Hovering uncertainty | Horizontal: ±1.5 m, Vertical: ±0.5 m |
| Position read-out uncertainty (with RTK) | Horizontal: ±1 cm, Vertical: ±2 cm |
| Orientation read-out uncertainty (with RTK) | ±0.8$^\circ$ |

product sheet DJI (2016)). This avoids developing a custom beam steering unit. The gimbal system can be used standalone and as an integrated part of the drone. It comes with its own IMU and a 32-bit Digital Signal Processor (DSP), which acts as a control unit for three servo motors equipped with encoders on their shafts. There are wired and wireless options when

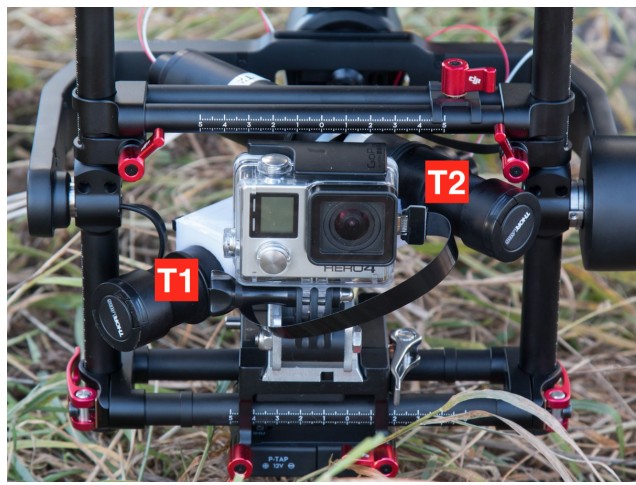

**Figure 2.** Gimbal payload: GoPro camera and two telescopes highlighted with T1 and T2 (here mounted with 90° opening angle between them) with protective lens caps. Picture taken from the photo gallery https://work.courtney.dk/#collection/7.

comes to the communication with the gimbal system. Like in the case of the drone, DJI provides a well-documented SDK and AP for the gimble system, allowing for example the implementation of custom motion profiles for all three axes either as feed rate (constant velocity) or time moves. Therefore, the gimbal system alone can act as a very good beam steering unit (i.e., scanner). Overall, the entire drone ecosystem (hardware, software, documentation, etc.), provides means for developers
to adapt the drone and gimbal for various applications. It should be pointed out that the specifications provided in Table 2 and 2 are acquired from DJI product sheets, and thus need to be validated.

To attach the two telescopes to the gimbal, we have 3D printed a mounting bracket for the telescopes. The bracket was made such that it allows manual setup of the 'opening' angle between telescopes. The bracket was attached to a custom-made aluminium 'tray' which slides into the gimbal camera mounting system. The tray also had a mount for a GoPro camera. The
entire custom-built sensor setup is shown in Figure 2.

## 3   Measurement campaigns description

As a part of the POC stage of the drone-based wind lidar system development we performed several measurement campaigns. In comparison to more typical wind measurement campaigns (e.g. Vasiljević et al., 2017) the POC stage campaigns were shorter in duration (10 to 30 min). With the POC campaigns our prime focus was to demonstrate the feasibility of the drone-based
wind lidar concept.

To operate the drone we followed the Danish drone rules set by Danish transport, construction and housing authority. Our drone pilots were required to have drone licences. The campaigns took place at the DTU Risø campus. The campus is located 5 km north of Roskilde, on the island of Zealand, Denmark . Within the campus there is a test center for wind turbines (Figure 3). The test center is surrounded by the Roskilde fjord (towards west), campus buildings (towards north), and agricultural land

**Table 3.** Basic specifications of gimbal from DJI product sheet (DJI, 2016)

| | |
|---|---|
| Dimensions | 280 mm (width) x 370 mm (length)x 340 mm (height) |
| Maximum dimensions of attached peripherals (recommended) | 160 mm (width) x 120 mm (depth) x 130 mm (height) |
| Weight (with vibration absorber) | 2.15 kg |
| Maximum payload weight | 4.5 kg |
| Runtime | 3 hours |
| Operating temperature | -15°C to 50°C |
| Maximum controlled angular velocity | Yaw axis: 200°/s, Pitch axis: 100°/s, Roll axis: 30°/s |
| Angular range | Yaw axis: endless, Pitch axis: +270° to -150°, Roll axis: ± 110° |
| Angular accuracy | ± 0.02° |

(towards east and south). It is located in flat terrain, though the terrain mildly slopes in the direction from the fjord towards the row of turbines. The prevailing wind direction is from the fjord, thus from the west. The test center includes test pads for small wind turbines (currently three in use) and several well-instrumented met masts. For the purpose of the drone campaigns we used measurements from two masts denoted as 'VT' mast and 'TW' mast in Figure 3. These two masts are 70 m and 10 m
tall respectively. The masts are IEC compliant and include calibrated in-situ wind sensors (cup or sonic anemometers and wind vanes). In this study we used measurements from a number of Metek USA-1 3D sonic anemometers. The flight conditions during the campaigns were good with relatively low wind speed and no precipitation (Table 4). The biggest challenge involved during the measurements was handling of the 100-m fibres to avoid snagging, which could lead to bend loss and disturbance of the drone's flight. Careful stacking of the uncoiled fibres on the ground was essential, with separation of the T (transmitting)
and R (receiving) fibres to minimise risk of tangling.

Instead of developing the drone/gimbal customization via SDK for the POC stage we decided initially to assess to what level the drone and gimbal straight "out of the box" are already a turn-key solution for wind lidar applications. Hence, we simply attached the telescopes to the gimbal system and manually positioned the drone and oriented the gimbal system using the on-board drone camera and GoPro camera mounted on the gimbal tray. The drone was manually steered to hover within
15 a couple of meters from the mast mounted sonic anemometers. These sensors were used as the reference for the comparisons in the text that will follow. The on-board drone camera was used to coarsely position the drone at the right height. Since the telescopes were attached to the gimbal, the GoPro camera was used to make sure that the telescopes are at the same height of the reference instrument and that they are pointing the laser beams in the desired directions. Once this was performed, simply the drone and gimbal were locked in the position. Afterwards, the drone A3 Pro flight controller automatically maintained the
20 drone position, and orientation and leveling of the telescopes carried by the gimbal. The focus of the telescopes and the opening angle between them were adjusted on the ground prior flights.

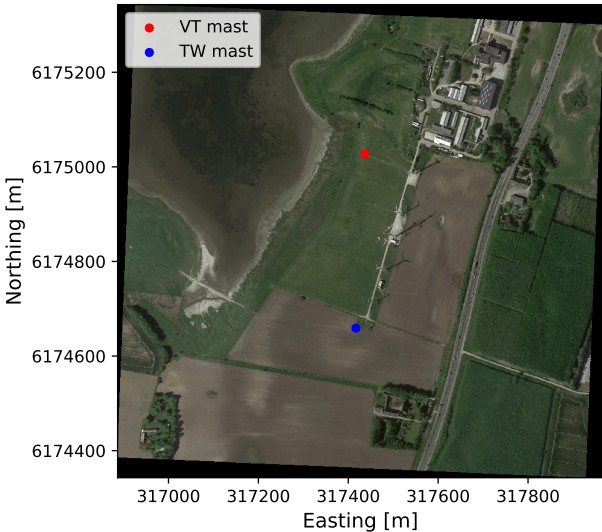

**Figure 3.** Aerial photo of test center at Risø campus: blue point - position of 10 m mast denoted 'TW', red point - position of 70 m mast denoted 'VT'. Aerial data: ©Google Maps, DigitalGlobe.

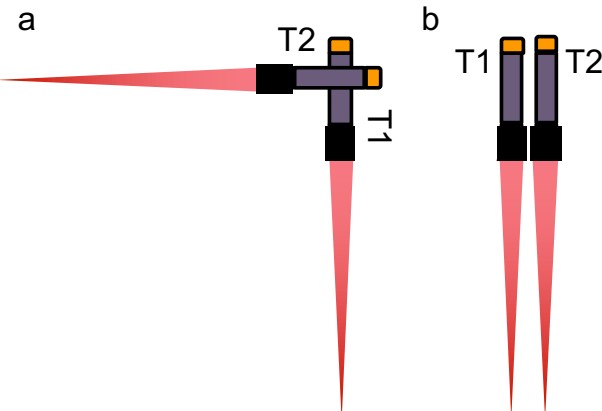

**Figure 4.** Telescope setup: a) opening angle of 90°, b) opening angle of 0°

We used two configurations of the opening angles (see Figure 4). In the first configuration the opening angle between telescopes was set to 90°, thus the outgoing beams were perpendicular to each other (Figure 4a). In the second configuration the opening angle was set to 0°, thus the outgoing laser beam were parallel to each other (Figure 4b).

We performed four experiments (See Table 4 with details). We started first with an experiment in which we hovered the drone in the vicinity of a sonic anemometer mounted 8-m agl on the 10-m mast (TW mast in Figure 3). With this experiment we secured the first batch of data for the inter-comparison between the mast and drone based wind sensors to prove the concept

**Table 4.** Overview of experiments

| Experiment no | Flight time [min] | Flight height [m] | Telescope orientation [deg] | Telescope setup [deg] | Wind speed [m/s] | Wind direction [deg] | Weather condition |
|---|---|---|---|---|---|---|---|
| 1 | 22 | 8 | 240 | 90 | 3 - 7 | 190 - 260 | Clear sky |
| 2 | 15 | 18, 31, 44, 57 and 70 | 20 | 90 | 2 - 6 | 206 - 259 | Clear sky |
| 3 | 16 | 70 | 20 | 90 | 3 - 6 | 206 - 251 | Clear sky |
| 4 | 6 x 5 | 5 | 30-70 | 0 | 1-4 | 30-70 | Cloudy |

feasibility. In the second experiment we hovered the drone at several heights next to the 70-m mast (VT mast in Figure 3) attempting to test the feasibility of doing multi-height measurements (i.e, vertical profiling of the wind) as well as acquiring the wind speed measurements beyond 60 m agl. In the third experiment, we explored the possibility of using measurements from a single telescope to reconstruct the horizontal wind speed since the single telescope concept is a lower cost option for drone-based wind lidar development. In these three campaigns we used the telescope configuration shown in Figure 4a, in which the focus distance of the two telescopes was fixed to 5 m to be sure that we are measuring the flow undisturbed by the drone presence. In the last experiment the goal was to detect the drone disturbance zone, i.e. the area of the air disturbed by the drone downwash (telescope configuration shown in Figure 4b). Although it would be logical to start with the downwash experiment, we first needed to get an assurance that the proposed drone-based lidar concept really works. Therefore we started directly by acquiring wind speed measurements close to a mast-mounted sensor with what later proved to be a rather conservative configuration of the focus distance.

### 3.1 Experiment 1: Hovering next to 10-m mast

To gain the experience in operating the newly-built drone-lidar system we started with measurements close to the TW mast. The 10-m mast was chosen for the first measurements as it does not have guy wires that could impose a risk while flying the drone close to it. We performed three flights next to this mast. During these flights the drone was positioned 6 m upstream of the sonic (8 m from the mast). We used the telescope configuration with the opening angle of $90°$ (see Figure 4a). This telescope configuration allowed us a straightforward measurements of the horizontal wind speed amplitude:

$$V_h = \sqrt{V_{LOS\_T1}^2 + V_{LOS\_T2}^2}, \tag{1}$$

where $V_h$ is the horizontal wind speed, while $V_{LOS\_T1}$ and $V_{LOS\_T2}$ are LOS speeds measured by the telescope 1 and 2 respectively.

Undoubtedly the previous relation is true assuming that:

  – the gimbal is capable of retaining the leveling of the telescope mount,

  – the flow is homogeneous at the two measured lidar focus points

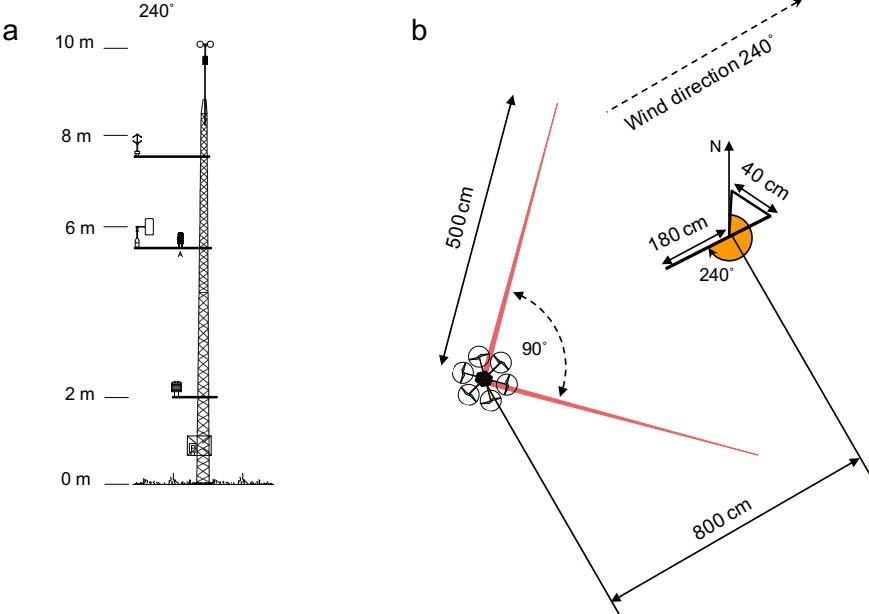

**Figure 5.** First experiment configuration: (a) 10-m mast sketch with position of instruments (2 m - humidity and temperature probe, 6 m - wind vane and humidity and temperature probe, 8 m - sonic anemometer, 10 m - cup anemometer ); (b) 10-m mast cross section at 8 m height with the orientation of the drone with respect to the boom at which the sonic anemometer is installed.

The focus distance as previously mentioned was set to 5 m, which considering the size of the optics results in the effective probe lengths of about 15 cm for each telescope. During the flights, using the gimbal system we were able to point the bisector of the LOS to face the sonic by pointing the beams downwind (see Figure 5). The wind direction during the flight was about 240°. As earlier mentioned with this configuration we measured the horizontal wind speed next to the sonic anemometer 5 mounted 8 m agl at the 10-m mast denoted TW in Figure 3.

The LOS velocity from the two telescopes and the horizontal wind speed measured with the sonic were collected with a 50 Hz data rate, though on two separate data acquisition system. To simplify the data sync and a follow up inter-comparison, both the sonic and drone data were averaged to give a 1 Hz sampling rate.

Figure 6 shows a time series and the correlation plot from the flight done next to the sonic anemometer mounted 8 m agl 10 at the TW mast. Considering that we do not account for the movement of the drone, which will influence the measured wind speed, the comparison is reasonably good.

### 3.2 Experiment 2: Hovering above several different heights next to V52 mast

Once we got confident in operating the drone-lidar system we started measurements next to the VT mast which has sonic anemometers installed at five heights (18 m, 31 m, 44 m, 57 m and 70 m agl see Figure 7a). With a total of 21 anchor points 15 distributed at seven heights the mast is fixed to the ground using corresponding number of guy wires, which required an extra

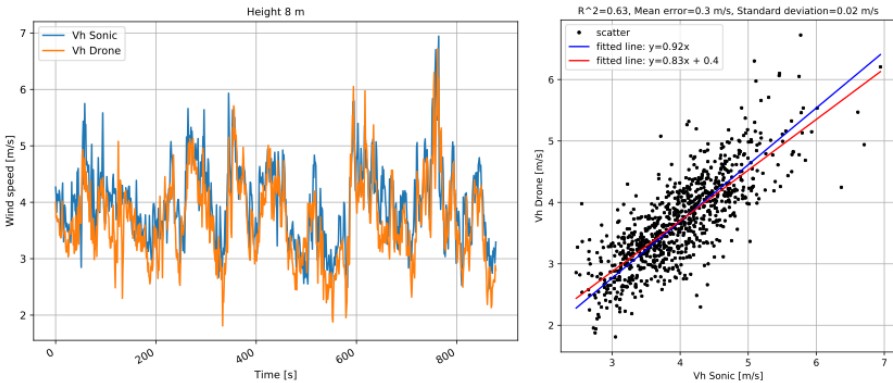

**Figure 6.** Measurements acquired next to the TW mast: left - 1 Hz time series of the horizontal wind speed acquired with the drone-lidar system and the sonic anemometer, right - linear regression plot.

caution during the drone flight. We have retained the opening angle of 90° between the two telescopes (see Figure 4a and Figure 7b). Using the gimbal system we were able to point the bisector of the LOS to face the sonic anemometers, however this time the beams were pointed into the wind (see Figure 7b). The wind direction during the flight was around 230°.

The aim of this experiment peformed next to the VT mast was to hover and measure wind speed next to each of the sonic anemometers (Figure 8), to vertically profile the wind. The total measurement time was approximately 15 minutes.

We started with the lowest sonic anemometer (18 m agl), measured for a period of time and then re-positioned the drone to the next height. This process continued until we reached the top mounted sonic anemometer. Once the measurements next to the top mounted sonic anemometer were completed we initiated the landing of the drone. During the measurement campaign the wind was coming from the southwest (mean wind direction of 230°). Accordingly, we positioned the drone the northeast of the mast, and oriented the gimbal system such that the laser beams were steered towards the southeast. In this way, the bisector of the two beams pointed approximately into the wind, thus the drone itself did not interfere with the flow where the beams were focused.

During the measurement campaign, we have manually started and stopped the measurements at each height. More accurately, we waited for the drone and gimbal operators to position the drone, and once this was completed we initiated measurements. Similarly, after a certain period of time we have stopped the measurements and indicated to the operators to move the drone to the consecutive height. At each height we produced two datasets which corresponded to the two telescopes.

Since we did not record the orientation of the gimbal system for each height we were able to determine only the amplitude of the horizontal wind speed. Like in case of the experiment next to the TW mast we averaged the 50 Hz data from the lidar and sonic anemometers to 1 Hz. Figure 9 shows 1 Hz data recorded by both sonic anemometers and drone-lidar system. Table 5 summarizes the results of all the comparisons.

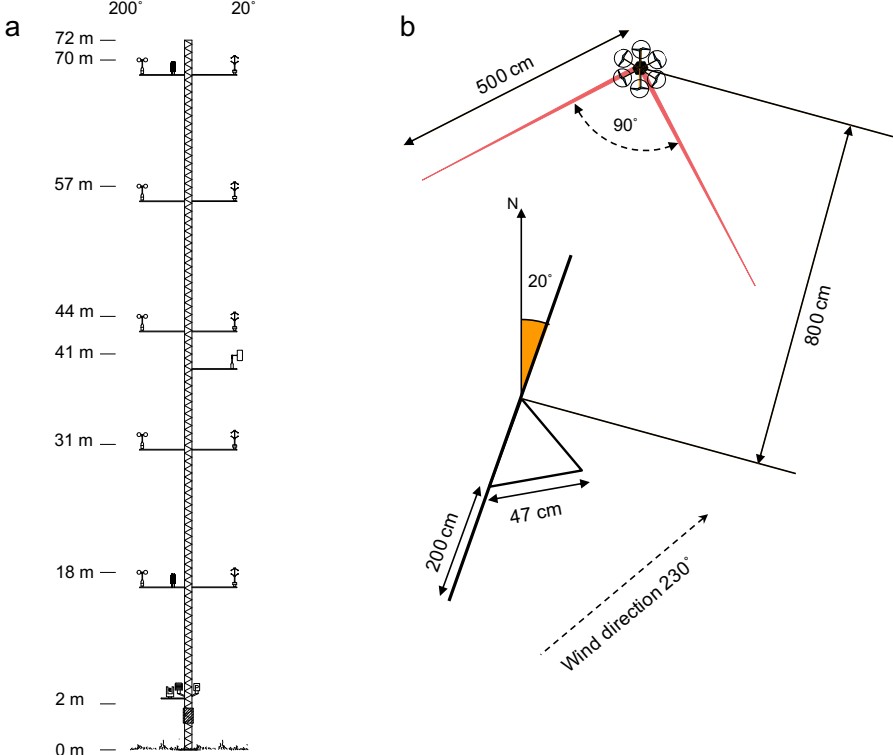

**Figure 7.** Second experiment configuration: (a) 70-m mast sketch with positions of instruments (2 m - humidity/temperature probe, 18 m - cup and sonic anemometers and humidity and temperature probe, 31 m - cup and sonic anemometers, 41 m - wind vane, 44 m - cup and sonic anemometers, 57 m - cup and sonic anemometers, 70 m - cup and sonic anemometers and humidity and temperature probe); (b) 70-m mast cross section at 70 m height with the orientation of the drone with respect to the boom at which the sonic anemometer is installed.

**Table 5.** Summary of comparison of multi-height measurements

| Parameters | 18 m | 31 m | 44 m | 57 m | 70 m |
|---|---|---|---|---|---|
| Number of samples | 157 | 104 | 89 | 108 | 323 |
| $R^2$ | 0.68 | 0.52 | 0.48 | 0.45 | 0.77 |
| Mean difference [m/s] | 0.08 | 0.05 | 0.06 | 0.02 | 0.11 |
| Standard deviation [m/s] | 0.04 | 0.06 | 0.08 | 0.06 | 0.03 |
| Slope | 1.01 | 0.98 | 0.98 | 0.99 | 0.97 |

From the aforementioned figure and table we can see that the best comparison between the measurements from the drone and mast is at the top height mainly due to the fact that we did spend most of the time hovering next to the corresponding sonic. Nevertheless, the comparisons are quite good for all heights considering that they are using high frequency (1Hz) data.

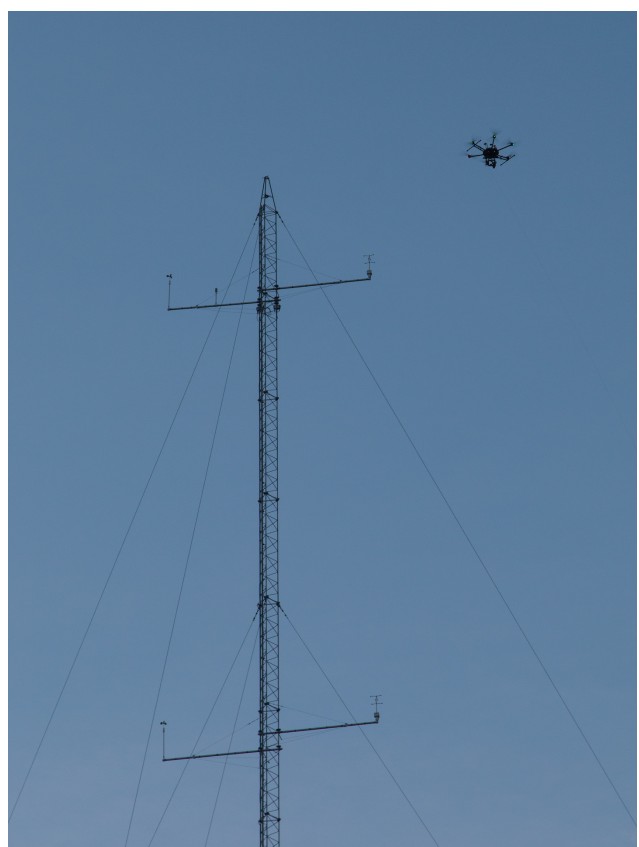

**Figure 8.** Drone parked next to sonic at 70 m agl

### 3.3  Experiment 3: Single-beam trial

The simplest, cheapest configuration for drone-mounted lidar is a single horizontally-aligned staring telescope. This is capable of measuring accurate wind speed if the beam is aligned closely to point along the wind direction. The resulting speed error is an underestimate given by the cosine of the misalignment angle. As such, the error is relatively small even for quite a significant misalignment (e.g., 1.5% for 10 degrees). This concept was tested in an experiment where the drone was hovered in close proximity to the sonic at 70 m, and only one of the telescope outputs (T1) was used for speed comparisons (Figure 10). It is likely that simple methods can be devised to align the beam with the wind based on drone flight characteristics, but in this experiment it was achieved by minimizing the Doppler offset observed by the other, orthogonal, telescope (T2). If this speed from T2 can be maintained close to zero, then this ensures the beam from T1 is closely aligned with the wind. Figure 10 shows the time series from analysis of approximately 15 minutes of data. When the speed measured by T2 is low, as in the first 250 seconds, then agreement between sonic and T1 is very good. Although the corresponding correlation plot from the full data period (Figure 10) exhibits significant scatter (a consequence of high data rate combined with drone-sonic separation of several

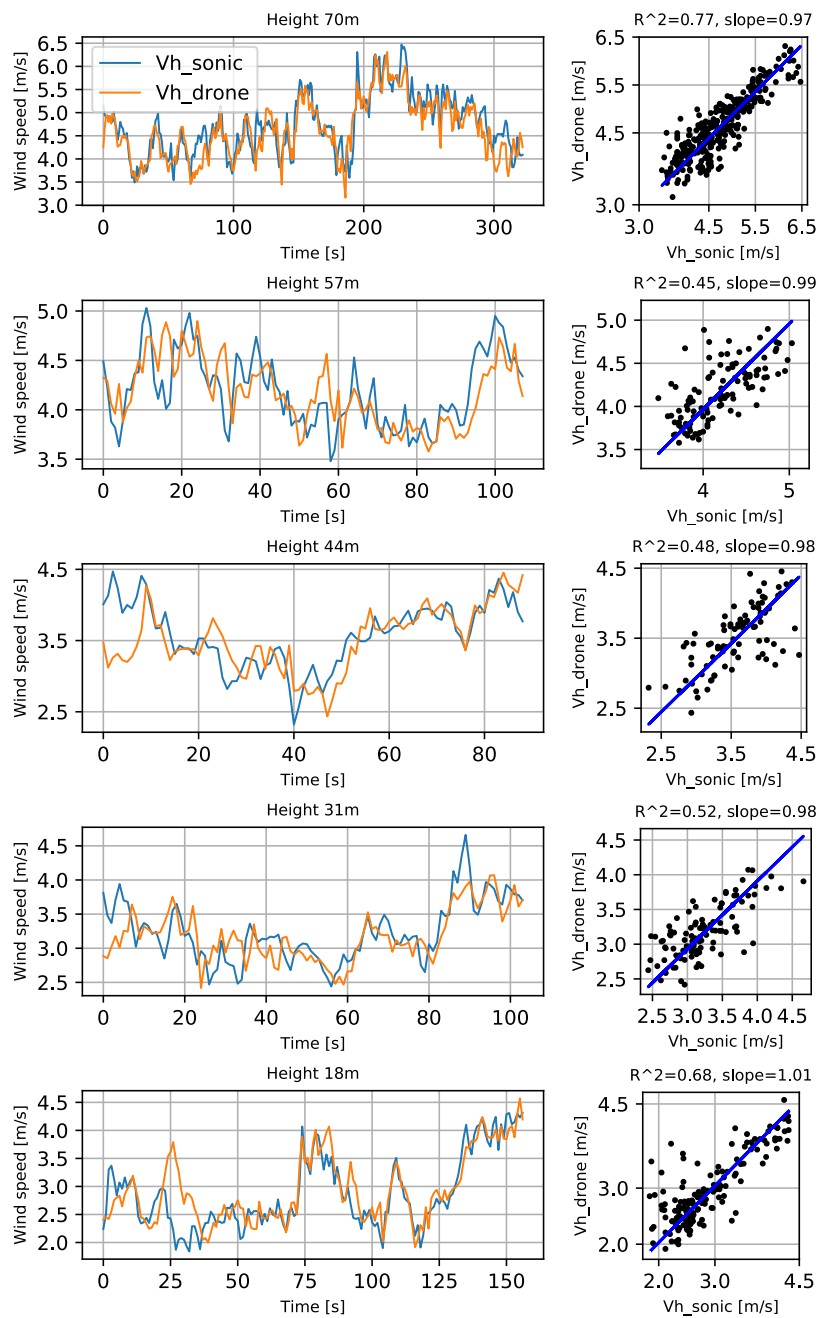

**Figure 9.** Comparison of multi-height measurements from the second experiment: left figures - time series, right figures - linear regression plot forced through zero

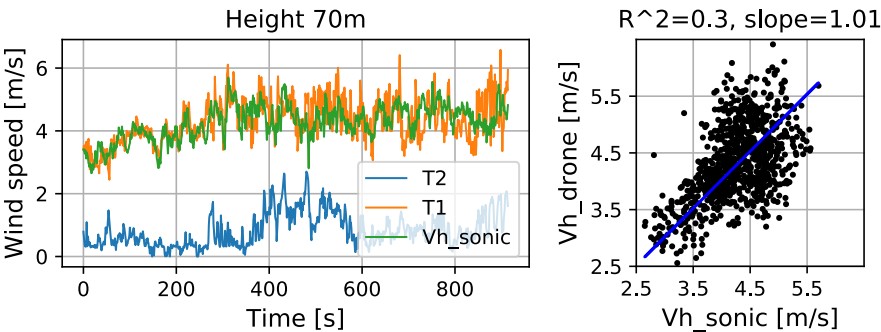

**Figure 10.** Single beam comparison at 70 m: left figure - time series, right figure - linear regression forced through ($R^2 = 0.3$, slope = 1.01, $\mu_{(Vh_{sonic}-T1)} = 0.05$ m/s, $\sigma_{(Vh_{sonic}-T1)}$=0.03 m/s)

metres), however, the mean wind speeds are in close agreement. The mean difference between the wind speed measured by sonic and T1 is 0.05 m/s with the standard deviation of difference of 0.03 m/s.

From this brief experiment we conclude that this approach could be a viable option as long as a reliable simple method of beam alignment along the wind direction can be devised.

## 3.4 Experiment 4: Drone disturbance zone

It is obvious that the downwash from the drone itself can severely influence the lidar measurements and it is therefore important to establish how far out from the drone centre this disturbance zone stretches. This was done by mounting the telescopes on top of each other in the gimbal with the laser beams pointing parallel and horizontally out from the drone (see Figure 4a). We first started by conservatively guessing that given the size of the drone at the distance of 5 m from its center the drone downwash should not have any impact on the free stream. Accordingly, in the first measurement series both telescopes were focused at 5 m and data acquired for about five minutes. The drone was then landed and the focus distance of telescope 2 decreased by approximately 1 m while that of telescope 1 was kept constant and a new measurement series acquired. We repeated this process six times until a focus distance 0.7 m for telescope 2 was reached.

The top image in Figure 11 shows an example of a 1 Hz time series obtained with telescope 2 focused at 2 m. With the measurement volumes being separated by only 3 m the measurements from telescope 1 and 2 are expected to resemble each other to a high degree and that is indeed seen to be case. Any differences between the two are mostly due to the measurement volumes not being exactly co-located. In contrast, the bottom image in Figure 11 shows a time series where telescope 2 is focused at 0.7 m with the measurement volume clearly within the drone disturbance zone. All though some general features are seen to be the same in the two plots telescope 2 measures a significantly lower average wind speed which is due to the blockage effect of the drone, while the downwash is perpendicular to the beam and hence has no Doppler contribution.

Figure 12 shows the relative difference in average wind speed measured by the two telescopes as function of telescope 2 focus distance. With the measurement volumes co-located at 5 m focus distance there is as expected virtually no difference

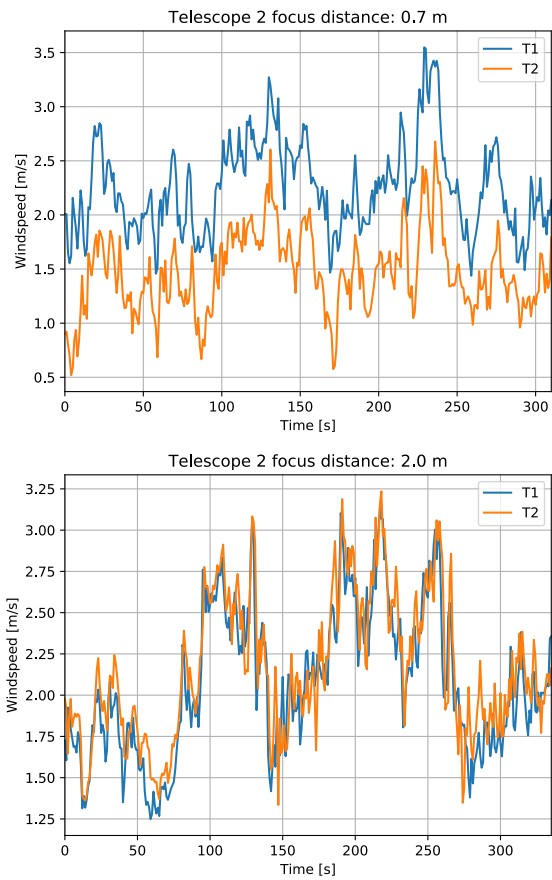

**Figure 11.** Example of 1-s averaged wind speed measurements acquired by the two telescopes: top image - telescope 1 focused at 5 m and telescope 2 at 2 m, bottom image - telescope 1 focused at 5 m and telescope 2 at 0.7 m.

between the measured average wind speeds and the difference stays low (less than $1\%$) down to 3 m. At 2 m focus distance there is a positive difference of about $3.9\%$ and at 1 m the difference becomes negative with a value of $-2.3\%$ indicating that the downwash starts to disturb the measurement. Finally, there is a very large difference of $-35\%$ when telescope 2 is focused at 0.7 m and thus measuring directly inside the disturbance zone.

5    From these measurements we conclude that the disturbance zone stretches between one and two metres from the drone centre, and when focusing the laser beams at three metres or more the influence of the drone itself on the measurements is negligible.

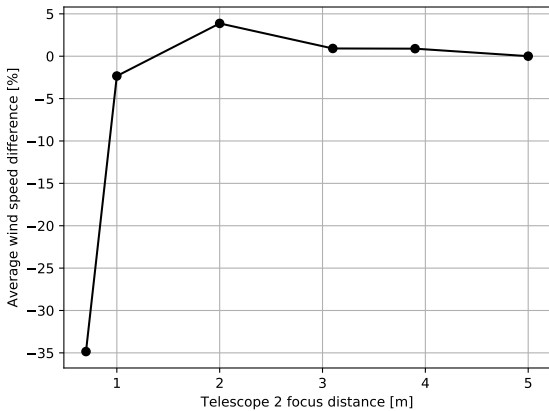

**Figure 12.** Relative difference in average wind speed measured by the two telescopes as function of telescope 2 focus distance.

## 4 Discussion

In our first two experiments (Section 3.1 and 3.2) we have endeavoured to demonstrate that the wind speed measured by the drone system is consistent with that measured by the nearby sonic anemometers. This has been successfully accomplished as can be seen from both the time series and scatter plots in Figures 6 and 9. A summary of the results from the VT mast is shown
in Table 5. Here we can see that the slope of the constrained linear regression is close to unity in all cases.

Whilst these tests serve to demonstrate the plausibility of the measured drone system wind speed, a more detailed and convincing testing methodology needs to be developed which can be used to assign a traceable uncertainty to the drone system. Two inherent problems are firstly the short flight time available and secondly the necessary separation between the drone system and the reference mast instrumentation. The latter gives rise to an inherent degradation in correlation at the time scales
available as dictated by the former (only a few minutes of flight time).

There are some obvious improvements to be made here before the method becomes applicable to practical real-world measurements - primarily maximising the flight time at any given height. Battery-powered drones cannot provide continuous measurements over long periods of many hours. Therefore, ultimately we envisage using a tethered system receiving flight power on its umbilical. Using such a system it should be possible to measure for hours as opposed to minutes. Alternatively,
a charging deck with several drones could be employed allowing near-uninterrupted measurements (a short interruption would probably occur during the drone substitute). However, even if we provide a continuous power to the drones, still the life expectancy of their motors is currently in the realm of couple of days of continuous operation after which the motors may need to be serviced or replaced. Nevertheless, the recent pace of development of drone technology suggests that we can expect significant improvements in their capabilities; this work aims to anticipate these developments that could make drone-mounted
lidar a practical and cheaper option in the future.

A second approach to assessing the uncertainty could be using what has become known as the 'white box' approach in the field of nacelle-mounted lidar calibration (Borraccino et al., 2016). For these systems it has been found that it is essentially impossible to compare the reconstructed wind speed directly with a reference measurement in a calibration environment. Instead the component uncertainties comprising the line-of-sight speed uncertainty and the uncertainty in the lidar geometry are propagated through the reconstruction algorithm to give an estimate of the uncertainty on the reconstructed speed. Drone-based lidar speed measurements lends itself to this technique for two reasons: Firstly the very short range lidars can be calibrated directly using a flywheel approach (Pedersen and Courtney, 2018) which has a much lower reference uncertainty than the alternative of using calibrated cup anemometers. Secondly the geometry and reconstruction is simple and easy to solve analytically.

Despite this being a first attempt we have already accomplished a close agreement to the nearby mast instrumentation. This has been achieved using only the basic drone position stabilisation and without correcting in any way for the drone motion. A clear next step (outlined below) is to implement the differential GPS positioning (already installed but inoperable in the initial tests) and to log this data. Not only will the drone be even more stable but we will be able to correct for the induced speed from the perturbations.

Our first experiments had the primary aim of demonstrating the recording of plausible wind speed from the drone system. Should the system later fail, our proof-of-concept would have been already been achieved. Actually the system proved to be quite robust (we had no catastrophic failures) and we were able to proceed with secondary but important tests. In the second experiment we have hovered the drone at multiple heights (mimicking mast measurements) which corresponded to the locations of sonic anemometers. In principle to measure wind profiles with the described system, a drone could vary position only in height (fixed Northing and Easting position of the drone), thus repetitively flying up and down between the range of heights. In the third experiment (Section 3.3) our aim was to investigate to what extent a single beam system would suffice to measure wind speed. Critical readers might with some justification claim that using two beams to show that only one is necessary is unconvincing. Potentially, we could substitute the second beam (here we used it to ensure that the cross-wind component was close to zero) with data from the drone (e.g., tilt or differential motor power) to substantiate that we are pointing into the wind. Alternatively, using the gimbal or drone itself a single beam could be steered in multiple directions performing a full or partial plan position indicator (PPI) scans and wind speed reconstructed from the acquired LOS measurements using know techniques such as Chen et al. (2017). Nevertheless, various methods for the substitution of the second beam should be tested to find the most suitable one.

In the last experiment (section 3.4) we tested our assumption that our initial focus distance of 5 m was outside the drone downwash influence zone. This was confirmed and it appears over-conservative for the set-up and conditions experienced here. We can conclude that at 3 m no downwash influence is discernible. In future testing we will use this as the focus distance. However, a dedicated study and modeling of the downwash under different conditions is necessary since we expect it to have different behaviour with respect to the payload and wind conditions.

Reducing the focus distance will reduce the separation between the probing beams. This is an advantage as the drone system will be able to measure in inhomogeneous flow with reduced error. In homogeneous flow, with a smaller probe separation, the drone system will be able to acquire meaningful time series of wind speed to a higher frequency since the flow will remain

correlated at the two probe positions over shorter averaging times. It will thus be possible to perform scalar averaging without significant error (this is a known problem with nacelle lidar systems).

Probe separation could be further reduced by using a smaller opening angle than the 90° used in this initial trial, or by separating the telescopes as much as possible and converging their beams. Conceivably the opening angle could be reduced to as little as 30° (common for nacelle-mounted lidars) especially if the drone tilt or differential rotor power can be used to keep the beams more or less aligned with the wind direction and away from the (for homodyne CW lidars) troublesome trans-zero zone where radial speed polarity issues (inability to distinguish between positive and negative) could also arise. Further work is clearly needed to find the optimal probe geometry for measuring horizontal as well as vertical wind speeds.

Conventionally, it has been troublesome for lidar to carry out point measurements of turbulence analogous to what a cup or sonic would measure: probe volumes extend typically over many metres, and the beams that interrogate the flow at different angles in a scanning system are often separated in space by many 10's of metres. By contrast, due to their short measurement range and very small measuring volume, drone lidar systems will be able to make turbulence measurements in and above the frequency range relevant in wind energy aplications. The lidars we have used have a probe length (FWHM) of 5 cm when focused at 3 m. This would be sufficient for useful wind speed spectral content up to at least 10 Hz at 10 m/s wind speeds. The lidar data acquisition is able to acquire signals at 50 Hz. The generation of accurate turbulence data from drone-mounted lidar requires knowledge of the telescope's orientation and speed to allow motion compensation, and this will form the topic for future study.

An obvious application is in-situ wake turbulence measurements where the drone can be positioned at a desired position relative to the turbine or perform a pre-described trajectory. Multiple drones (swarms) could also be envisioned to provide simultaneous measurements at a number of positions.

Similarly a drone swarm upstream of a wind turbine could provide inflow data of unprecedented detail and quality for power performance or load validations, including horizontal and vertical shear, wind veer, turbulence intensities and also the spatial structure (coherence) of (at least) the longitudinal turbulence component.

Many other drone-lidar applications can be envisioned once some degree of drone autonomy can be developed (ability to fly pre-programmed sequences, land and re-charge automatically, fly again,...). Indeed, the power and load verification duties described above only become realistic outside a research environment once these abilities are developed. Truly operational applications could include flow monitoring inside and upstream of wind farms. This could both enhance wind farm control and provide a degree of forecasting. Outside wind energy there are obvious applications in wind engineering (e.g. flow, turbulence and coherence measurements at remote sites) as well as exciting possibilities in a variety of recreational areas (sailing, golfing, ski-sport).

## 5   Conclusion

A novel wind measurement technique based on the fusion of a standard drone with a prototype wind lidar has been reported. We have described the proof of concept (POC) drone-lidar system which was developed to demonstrate the feasibility of

this new measurement technique. Besides the POC system description, we have reported on the first experiments performed with this system. In these experiments the drone-disturbance zone (caused largely by the influence of drone downwash) was characterized, and inter-comparison was performed of wind measurements acquired by the drone-based wind lidar with those acquired by adjacent mast-mounted sonic anemometers. A good agreement between the sonic anemometers and the drone-based wind lidar measurements has been found even without any motion compensation.

It is expected that motion compensation will result in further improvement of wind speed accuracy, and will also allow detailed investigation of turbulence. The rapid data rates and very small measurement volume suggests the exciting prospect of tracking fast atmospheric fluctuations, and deriving turbulence spectra for comparison with cup and sonic anemometers.

Overall, excellent results have been obtained in these first attempts with a fairly simple measurement system which provide a necessary vindication for the proposed measurement technique and secure the foundation for the technique's future developments.

*Author contributions.* Conceptualization, N.V., M.H. and M.C.; Methodology, N.V., M.H., M.C., A.T.P., M.P., J.H. and K.B.; Software, M.P. and M.H.; Validation, N.V., M.H., M.C. and A.T.P.; Formal Analysis, N.V., M.H., A.T.P. and G.R.T.; Investigation, N.V., M.H., A.T.P. and G.R.T.; Resources, N.V., M.H., M.P. and J.H.; Data Curation, N.V., G.R.T., M.P., J.H. and K.B.; Writing - Original Draft, N.V., M.H., A.T.P., G.R.T. and M.C.; Writing - Review & Editing, N.V., M.H. and M.C.; Visualization, N.V., A.T.P., G.R.T. and M.H.;Supervision, N.V. and M.H.; Project Administration, N.V. and M.C.; Funding Acquisition, N.V. and M.C.

*Competing interests.* Michael Harris and Mark Pitter are employed at ZX Lidars which is a company focused on the commercialization of the wind lidar technology. Other authors declare that they have no conflict of interest.

*Acknowledgements.* The authors would like to express special gratitude to Michael Rassmussen (DTU), Claus Pedersen (DTU) and Per Hansen (DTU), for their help in setting up and operating the measurement system, and Poul Hummelshøj (DTU) for backing up the internal project.

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
