# Peer review of "Wind sensing with drone mounted wind lidars: proof of concept"

_Atmospheric Measurement Techniques, 2019_

## Referee Comment (RC1) · Anonymous Referee #2 · 18 Aug 2019

**1   General comments**

In the manuscript by Vasiljevic et al. a proof of concept is presented for a drone-mounted lidar for wind measurements. This concept is an important development in wind measurement technology and opens a lot of possibilities for scientific applications. The results that are presented are very promising and are definitely worth publishing in the journal "Atmospheric Measurement Techniques". I have some concerns about the presentation of the results and requests for changes which the authors should consider before submitting a revised manuscript:

[Figure]

- Drone-mounted lidar measurements are introduced in this manuscript as a possible replacement for ground-based lidar measurements or meteorological masts. I am very skeptical if this is a realistic scenario, given the logistical constraints that drone measurements have. With this I mean flight time, regulations, operation in all weather conditions (what are maximum wind speeds for example? What about rain?). The authors mention in the conclusion that a lot more development needs to happen before drones can be a serious alternative for power and load verification. However, I think that even with the present research version of this system, a lot of important measurements towards validation of ground-based lidars, studying of wake dynamics and especially turbulence research can be done. I think these applications should be emphasized much more. For this purpose, I wish to see a comparison of turbulence spectra between sonic anemometers and drone lidars for the presented measurements.

- The description of the state of the art is very weak on external references to both, drone-based measurements and short-range lidars. Most of the references are DTU-internal, although a lot of work is done world-wide on Doppler wind lidars. What are other CW-lidar systems and how is the used system different. Drone measurements are the topic of many research groups within the ISARRA community. It would be good to evaluate the presented system to other concepts which exist internationally (for example sonic anemometers or flow probes on drones). I want to urge the authors to improve their literature research and give credit to other researchers working in the field.

- A major concern for many drone-users are regulations and flight permissions. It would be great if the conditions for flights at the Riso campus could be explained in the experiment section.

- More specific comment are given below.

[Figure]

**2  Specific comments**

**2.1  Abstract**

*p.1,l.1*: The authors mention substiantially lower costs at the very beginning of the abstract and thus the manuscript, but do not mention the costs of the presented system at all. Can they make any comment on the costs? If not, please remove the statement.
*p.1,l.8*: I suggest to make the statement "Generally, very good agreement was found" a little more specific.

**2.2  Introduction**

*p.2,l.20*: Maybe any kind of precipitation should be added to problematic conditions for lidar measurements.
*p.2,l.9*: Give references for CW-lidars for wind energy research or atmospheric research in general. Same for pulsed lidars.
*p.3,l.11*: The explanation that the AOM can be eliminated should go into the system description in Sect. 2. A literature review of other methods to sense wind with drones would be very adequate in the introduction. There are a lot more references than Brosy et al. 2017.
*p.3,l.20*: Are measurements above thick forests with a drone lidar with fibre connection to the ground really realistic?

**2.3   Section 2**

*p.4-7, Fig. 1-3*: Instead of pictures of the single components, it would be good to have a picture with the lidars mounted on the drone gimbal.

*p.3, l.8*: What does 100% duty cycle mean here?

*p.5, Tab. 1*: Please add lidar wavelength and weight of the telescopes, as well as the focal range that can be adjusted.

*p.5, l.9*: "... with payload." How big is the payload?

*p.5, ll.10ff*: If I understand correctly the system has three GNSS-systems, plus another DGNSS system. This seems quite excessive. Isn't the RTK-DGNSS-system more than enough? Real-time DGNSS is not a synonym for RTK. RTK needs a feed of correction data from a nearby base station (not on the rover) to achieve the cm-accuracy. Is this really given in the setup?   I think the most important feature are the two antennas for improved pitch and yaw estimation.   With the short baseline of only 25 cm it is however quite questionable if this information really improves the IMU-information. Has this been evaluated? Please be momre specific in this explanation and distinguish between a dual-antenna setup with a baseline between the two antennas on the rover and an RTK setup with a baseline between base and rover that can be up to a few kilometers and I am not sure if this is really given here.   If not (as suggested in the discussion), it is not relevant in this study and should not be "advertised" here.

*p.7, Fig.3*: Please indicate the lidar telescopes. Since they look much different than in Fig.1, this is quite confusing. I suggest to remove Fig. 1 and label Fig. 3 accordingly.

*Section 2*: How big is the actual take-off weight of the system in this configuration?

**2.4 Section 3**

*p.8, ll.1-14*: I think it does not really add much to the study to present the preliminary tests with yet another system that has not been fully described, so I suggest to just skip this test and only present the results of the POC system introduced in Sect. 2. Figure 4 could be removed in my opinion as well because there is no reference instrument or further evaluation.

*p.8, l.22*: Does 'VT' and 'TW' have any specific meaning?

*p.9, l.1*: Please mention what kind of sonics are mounted.

*p.9, l.2f*: I am not sure if I understand the positioning procedure. What does it mean that adjustments were done using the GoPro camera?

*p.10*: I would suggest to make a list of flights that are analyze in this study, with flight time, flight height, telescope orientation and wind and weather conditions.

*p.11, l.12*: Giving a probed area of 7.07 m x 0.15 m seems a bit awkward and confusing. Maybe it would be easier to understand if you say that homogeneity at the two measured lidar focus points needs to be assumed.

*p.13, ll.11ff*: I think this conclusion can not be drawn. The higher $R^2$-value is probably due to the higher number of samples at 70 m. The standard deviation is mostly a measure for turbulence in this case, and in fact the slope of the linear regression is furthest from one. I doubt if a statistical analysis of these very short measurement periods makes any sense at all. I would suggest to show the time series and a comparison of the vertical profiles of tower and drone lidar, including the standard deviation as error bars and quantifying the mean difference instead of the linear regression and point clouds.

*p.15, Figs. 11-15*: I would suggest to combine these plots in some way. It is probably not necessary to show the single telescope radial wind speeds. I do also not think that the point clouds and linear regressions are necessary and even statistically significant. So, maybe just show the comparison of time series of drone lidar horizontal wind compared to sonic anemometers on all heights in one plot with subplots.

**2.5 Section 4**

*p.20, l.11f*: I actually disagree and think this is a very good way to show the influence of the out-of-line wind component on the reconstruction from a single beam. I do not think this could be done any better with drone-internal data which is subject to many other uncertainties. However, a comparison of these drone-internal methods with the lidar would be very interesting in future, of course.

*p.20, ll.16ff*: Has any indication been found that the downwash distance depends on wind direction and wind speed? If not, this could be a field of further research and a reason to be more conservative in future tests as well.

*p.21, ll.23ff*: I would recommend to not describe technical details of possible future developments in a scientific paper. Depending on how developments go, concepts might have to be adapted and it is not really relevant for the scientific ideas and visions.

**3 Technical corrections**

*p.2,l.15*: Use the proper
texteurosign instead of EURO.

*p.3,l.23*: The abbreviation POC is only introduced in the abstract, but should be introduced in the main text as well.

*p.4,l.5*: There is something wrong here: what is a "demonstrator for an nairborne wind speed requirement"?

*p.20,l.9*: Some grammar issue here.

---

## Referee Comment (RC2) · Anonymous Referee #3 · 23 Sep 2019

Great paper!

p 2-3: the list of criteria, should, at least for EPAs, also include a long duration (months) of the measurements, as a requirement Table 1: weight of telescopes?

Fig 7b: I don't understand why they have been put directly in front of each other? Potential flow distortion?

Discussion: How to measure profiles?

other ways of using a drone to measure the wind speed (eg differential motor power)

---

## Author Comment (AC1) · 28 Nov 2019

**1 General comments**

In the manuscript by Vasiljevic et al. a proof of concept is presented for a drone-mounted lidar for wind measurements. This concept is an important development in wind measurement technology and opens a lot of possibilities for scientific applications. The results that are presented are very promising and are definitely worth publishing in the journal "Atmospheric Measurement Techniques". I have some concerns about the presentation of the results and requests for changes which the authors should consider before submitting a revised manuscript.

*Dear referee,*

*We would like to thank you for the positive review of our paper.*
*We have incorporated your suggestions and made changes to our manuscript.*

*We provide point-by-point responses (in blue) together with the attached revised manuscript which contains highlighted differences between the original and revised manuscript (latexdiff was used for this task). However, we must notice that the older version of our manuscript has been reviewed and not the latest one. The difference between the first and second version are not major, however, there are some differences (e.g., the second version has improved Discussion, figures representing measurement setup, grammar issues fixed, EURO replaced by €, etc.).*

*Besides the revised manuscript which contains the highlighted differences we are also submitting separately the revised manuscript without the highlighted differences.*

Drone-mounted lidar measurements are introduced in this manuscript as a possible replacement for ground-based lidar measurements or meteorological masts. I am very skeptical if this is a realistic scenario, given the logistical constraints that drone measurements have. With this I mean flight time, regulations, operation in all weather conditions (what are maximum wind speeds for example? What about rain?). The authors mention in the conclusion that a lot more development needs to happen before drones can be a serious alternative for power and load verification. However, I think that even with the present research version of this system, a lot of important measurements towards validation of ground-based lidars, studying of wake dynamics and especially turbulence research can be done. I think these applications should be emphasized much more. For this purpose, I wish to see a comparison of turbulence spectra between sonic anemometers and drone lidars for the presented measurements.

*Currently, the drone-based wind lidars cannot replace met masts. However, considering the ongoing developments in drone technology (which is not driven by wind energy or science but due to the need of for example cheaper way of delivering goods) there is potential that this becomes realistic scenario.*
*The turbulence analysis, indeed of high importance, is not the scope of this paper primarily since the described measurement setup is not suitable to acquired necessary data (e.g., missing accurate information of the gimbal orientation). Nevertheless, it will be investigated in future.*

The description of the state of the art is very weak on external references to both, drone-based measurements and short-range lidars. Most of the references are DTU-internal, although a lot of work is done world-wide on Doppler wind lidars. What are other CW-lidar systems and how is the used system different. Drone measurements are the topic of many research groups within the ISARRA community. It would be good to evaluate the presented system to other concepts which exist internationally (for example sonic anemometers or flow probes on drones). I want to urge the authors to improve their literature research and give credit to other researchers working in the field.

*In the revised manuscript, we provide an overview of sUAS measurements of wind speed also the list of references related to the wind lidar measurements in wind energy domain has been extended.*

A major concern for many drone-users are regulations and flight permissions. It would be great if the conditions for flights at the Riso campus could be explained in the experiment section.

*In the revised manuscript we have stated that we operated the drone in accordance to the Danish drone rules. Also we added weather conditions during the flights.*

**2 Specific comments**
**2.1 Abstract**
p.1,l.1: The authors mention substantially lower costs at the very beginning of the abstract and thus the manuscript, but do not mention the costs of the presented system at all. Can they make any comment on the costs? If not, please remove the statement.
*Since at this stage we cannot provide the exact reduction in costs of such a system the statement was removed.*

p.1,l.8: I suggest to make the statement "Generally, very good agreement was found" a little more specific.
*The stated sentence was replaced with the following one:*
*"On average, an agreement to about 0.1 m/s between mast- and drone- based measurements of the horizontal wind speed was found"*

**2.2 Introduction**
p.2,l.20: Maybe any kind of precipitation should be added to problematic conditions for lidar measurements.
*The following sentence was added to the revised manuscript:*
*"Furthermore, any precipitation will affect the wind speed measurements by lidars. Specifically the vertical component of the wind will be biased since the lidar will dominantly measure the fall velocity of the precipitation (e.g., rain droplets)."*

p.2,l.9: Give references for CW-lidars for wind energy research or atmospheric research in general. Same for pulsed lidars.
*A number of references targeting specific wind energy applications has been cited in Introduction of the revised manuscript.*

p.3,l.11: The explanation that the AOM can be eliminated should go into the system description in Sect. 2. A literature review of other methods to sense wind with drones would be very adequate in the introduction. There are a lot more references than Brosy et al. 2017.
*We have revised the manuscript accordingly. The AOM explanation is moved to Section 2, while the Introduction contains a review of different methods of sensing wind with drones and fixed-wing sUAS.*

p.3,l.20: Are measurements above thick forests with a drone lidar with fibre connection to the ground really realistic?
*Due to the indicating location in the manuscript for this comment it seems that the reviewer did not review the second version of our manuscript since the indicated statement appears in the second version of manuscript at p.3, l27. Nevertheless, it is realistic to do such measurements with non-tethered drone which is now stated in the indicated sentence.*

**2.3 Section 2**

p.4-7, Fig. 1-3: Instead of pictures of the single components, it would be good to have a picture with the lidars mounted on the drone gimbal.
*FIgure 2 and Figure 3 show two telescopes mounted on the drone gimbal.*

p.3, l.8: What does 100% duty cycle mean here?
*It means that the laser beams are simultaneously lased through the two telescopes, thus that measurements are performed on two channels continuously and simultaneously (which is stated in the manuscript). As it is not bringinging any additional information '100% duty cycle' has been removed from the manuscript. We are confirming that the reviewer reviewed the first version of the manuscript due to the indicated page and line number of the comment.*

p.5, Tab. 1: Please add lidar wavelength and weight of the telescopes, as well as the focal range that can be adjusted.
*We have added requested information to the revised manuscript.*

p.5, l.9: "... with payload." How big is the payload?
*The stated flight time is with a payload of 5 kg. This information is now enclosed in the text of the manuscript.*

p.5, ll.10ff: If I understand correctly the system has three GNSS-systems, plus another DGNSS system. This seems quite excessive. Isn't the RTK-DGNSS-system more than enough? Real-time DGNSS is not a synonym for RTK. RTK needs a feed of correction data from a nearby base station (not on the rover) to achieve the cm-accuracy. Is this really given in the setup? I think the most important feature are the two antennas for improved pitch and yaw estimation. With the short baseline of only 25 cm it is however quite questionable if this information really improves the IMU-information. Has this been evaluated? Please be more specific in this explanation and distinguish between a dual-antenna setup with a baseline between the two antennas on the rover and an RTK setup with a baseline between base and rover that can be up to a few kilometers and I am not sure if this is really given here. If not (as suggested in the discussion), it is not relevant in this study and should not be "advertised" here.
*The system by default comes with three GNSS-system antennas. The additional two are antennas are part of so-called D-RTK system (differential GPS measurements + RTK) which beside the two additional antennas contains local mobile ground based RTK station (i.e., rover). In our tests we only used differential measurements and not RTK due to some issues in setting up the local rover. Also, the stated accuracy and other drone or gimbal specifications are provided in DJI product sheets which are now refered to.*

p.7, Fig.3: Please indicate the lidar telescopes. Since they look much different than in Fig.1, this is quite confusing. I suggest to remove Fig. 1 and label Fig. 3 accordingly.
*Figure 1 was removed, the telescopes are highlighted on Fig3 (now Fig 2).*

Section 2: How big is the actual take-off weight of the system in this configuration?
*About 3 kg (gimbal + gimbal payload(telescopes, GoPRo camera) + fiber cables )*

**2.4 Section 3**
p.8, ll.1-14: I think it does not really add much to the study to present the preliminary tests with yet another system that has not been fully described, so I suggest to just skip this test and only present the results of the POC system introduced in Sect.2. Figure 4 could be removed in my opinion as well because there is no reference instrument or further evaluation.
*We agree with the reviewer and accordingly we have removed the part of the manuscript related to the preliminary tests.*

p.8, l.22: Does 'VT' and 'TW' have any specific meaning?
*VT stands for V52 Tower (VT), while TW stands for True Wind (TW).*

p.9, l.1: Please mention what kind of sonics are mounted.
*Metek USA-1 3D sonic anemometers*

p.9, l.2f: I am not sure if I understand the positioning procedure. What does it mean that adjustments were done using the GoPro camera?

*DJI Drone has built in camera. However the telescopes were attached to the gimbal which can 'freely' move with respect to the drone. Therefore, we used the drone camera to roughly know that we are close to the reference instrument, while the GoPro camera (which was attached at the tray which carried the telescopes) was used to make sure that we are at the same height as the reference instrument and that we are pointing the laser beams in a right direction (avoiding the beams hitting the mast). We enclosed this clarification in the revised manuscript.*

p.10: I would suggest to make a list of flights that are analyze in this study, with flight time, flight height, telescope orientation and wind and weather conditions.
*New table has been added to the manuscript.*

p.11, l.12: Giving a probed area of 7.07 m x 0.15 m seems a bit awkward and confusing. Maybe it would be easier to understand if you say that homogeneity at the two measured lidar focus points needs to be assumed.
*We agree with the reviewer.*

p.13, ll.11ff: I think this conclusion can not be drawn. The higher $R^2$-value is probably due to the higher number of samples at 70 m. The standard deviation is mostly a measure for turbulence in this case, and in fact the slope of the linear regression is furthest from one. I doubt if a statistical analysis of these very short measurement periods makes any sense at all. I would suggest to show the time series and a comparison of the vertical profiles of tower and drone lidar, including the standard deviation as error bars and quantifying the mean difference instead of the linear regression and point clouds.
*We reformulate the statement regarding why the best comparison is at 70 m according to the reviewer comment. Regarding the presentation of the results in plots we prefer to keep our approach. Besides the linear regression plots we do indeed show the mean difference between the sonic- and drone- based measurements (see Table 4 in the reviewed manuscript).*

p.15, Figs. 11-15: I would suggest to combine these plots in some way. It is probably not necessary to show the single telescope radial wind speeds. I do also not think that the point clouds and linear regressions are necessary and even statistically significant. So, maybe just show the comparison of time series of drone lidar horizontal wind compared to sonic anemometers on all heights in one plot with subplots.
*The indicated figures are combined as a single plot and radial velocities from individual telescopes are removed from plots. We prefer keeping the linear regression plots since they are more indicative on the quantitative aspect of the measurement accuracy, while the time series plots are more useful for qualitative analysis.*

**2.5 Section 4**
p.20, l.11f: I actually disagree and think this is a very good way to show the influence of the out-of-line wind component on the reconstruction from a single beam. I do not think this could be done any better with drone-internal data which is subject to many other uncertainties. However, a comparison of these drone-internal methods with the lidar would be very interesting in future, of course.
*We improved that part of manuscript stating that various methods should be tested to find a suitable one for single beam applications.*

p.20, ll.16ff: Has any indication been found that the downwash distance depends on wind direction and wind speed? If not, this could be a field of further research and areas on to be more conservative in future tests as well.
*We did not investigate the downwash behaviour under different wind conditions. We agree that the future studies should include a dedicated research on the behaviour of the downwash under different conditions (wind speed, wind direction, payload, etc.). We have highlighted this in the revised manuscript.*

p.21, ll.23ff: I would recommend to not describe technical details of possible future developments in a scientific paper. Depending on how developments go, concepts might have to be adapted and it is not really relevant for the scientific ideas and visions

*The reviewer refers to the first version of the manuscript which undergone a brief assessment whether it is suitable for the AMT or not. After that first version of the manuscript we have submitted the second version, which actually does not contain the technical details on the future development.*

**3 Technical corrections**

p.2,l.15: Use the proper text euro sign instead of EURO.

*Similar like the previous comment. i.e. this was already changed in the second version of our manuscript. Therefore, the revised manuscript (version three) also has the correct sign.*

p.3,l.23: The abbreviation POC is only introduced in the abstract, but should be introduced in the main text as well.

*We followed the reviewer recommendation.*

p.4,l.5: There is something wrong here: what is a "demonstrator for an nairborne windspeed requirement"?

*We have removed the indicated part of the sentence.*

p.20,l.9: Some grammar issue here.

*The manuscript was revised accordingly.*

---

## Author Comment (AC2) · 28 Nov 2019

Our responses are provided as a suplementary PDF file due to highlighting of point-to-point responses.

Please also note the supplement to this comment:
https://www.atmos-meas-tech-discuss.net/amt-2019-102/amt-2019-102-AC2-supplement.pdf

---

## Editor Decision (ED1)

Please make the indicated changes in my marked up manuscript, which are mostly wordsmithing.

Note on page 2, line 23 you list a single paper (which I don't think is a review) as a "variety of atmospheric studies." Please find a more recent review paper of the field and cite that, or find several examples of different and more recent scientific studies and cite those.

---

## Author Response (AR3)

**1 General comments**

In the manuscript by Vasiljevic et al. a proof of concept is presented for a drone-mounted lidar for wind measurements. This concept is an important development in wind measurement technology and opens a lot of possibilities for scientific applications. The results that are presented are very promising and are definitely worth publishing in the journal "Atmospheric Measurement Techniques". I have some concerns about the presentation of the results and requests for changes which the authors should consider before submitting a revised manuscript.

*Dear referee,*

*We would like to thank you for the positive review of our paper.*
*We have incorporated your suggestions and made changes to our manuscript.*

*We provide point-by-point responses (in blue) together with the attached revised manuscript which contains highlighted differences between the original and revised manuscript (latexdiff was used for this task). However, we must notice that the older version of our manuscript has been reviewed and not the latest one. The difference between the first and second version are not major, however, there are some differences (e.g., the second version has improved Discussion, figures representing measurement setup, grammar issues fixed, EURO replaced by €, etc.).*

*Besides the revised manuscript which contains the highlighted differences we are also submitting separately the revised manuscript without the highlighted differences.*

Drone-mounted lidar measurements are introduced in this manuscript as a possible replacement for ground-based lidar measurements or meteorological masts. I am very skeptical if this is a realistic scenario, given the logistical constraints that drone measurements have. With this I mean flight time, regulations, operation in all weather conditions (what are maximum wind speeds for example? What about rain?). The authors mention in the conclusion that a lot more development needs to happen before drones can be a serious alternative for power and load verification. However, I think that even with the present research version of this system, a lot of important measurements towards validation of ground-based lidars, studying of wake dynamics and especially turbulence research can be done. I think these applications should be emphasized much more. For this purpose, I wish to see a comparison of turbulence spectra between sonic anemometers and drone lidars for the presented measurements.

*Currently, the drone-based wind lidars cannot replace met masts. However, considering the ongoing developments in drone technology (which is not driven by wind energy or science but due to the need of for example cheaper way of delivering goods) there is potential that this becomes realistic scenario.*
*The turbulence analysis, indeed of high importance, is not the scope of this paper primarily since the described measurement setup is not suitable to acquired necessary data (e.g., missing accurate information of the gimbal orientation). Nevertheless, it will be investigated in future.*

The description of the state of the art is very weak on external references to both, drone-based measurements and short-range lidars. Most of the references are DTU-internal, although a lot of work is done world-wide on Doppler wind lidars. What are other CW-lidar systems and how is the used system different. Drone measurements are the topic of many research groups within the ISARRA community. It would be good to evaluate the presented system to other concepts which exist internationally (for example sonic anemometers or flow probes on drones). I want to urge the authors to improve their literature research and give credit to other researchers working in the field.

*In the revised manuscript, we provide an overview of sUAS measurements of wind speed also the list of references related to the wind lidar measurements in wind energy domain has been extended.*

A major concern for many drone-users are regulations and flight permissions. It would be great if the conditions for flights at the Riso campus could be explained in the experiment section.

*In the revised manuscript we have stated that we operated the drone in accordance to the Danish drone rules. Also we added weather conditions during the flights.*

**2 Specific comments**
**2.1 Abstract**
p.1,l.1: The authors mention substantially lower costs at the very beginning of the abstract and thus the manuscript, but do not mention the costs of the presented system at all. Can they make any comment on the costs? If not, please remove the statement.
*Since at this stage we cannot provide the exact reduction in costs of such a system the statement was removed.*

p.1,l.8: I suggest to make the statement "Generally, very good agreement was found" a little more specific.
*The stated sentence was replaced with the following one:*
*"On average, an agreement to about 0.1 m/s between mast- and drone- based measurements of the horizontal wind speed was found"*

**2.2 Introduction**
p.2,l.20: Maybe any kind of precipitation should be added to problematic conditions for lidar measurements.
*The following sentence was added to the revised manuscript:*
*"Furthermore, any precipitation will affect the wind speed measurements by lidars. Specifically the vertical component of the wind will be biased since the lidar will dominantly measure the fall velocity of the precipitation (e.g., rain droplets)."*

p.2,l.9: Give references for CW-lidars for wind energy research or atmospheric research in general. Same for pulsed lidars.
*A number of references targeting specific wind energy applications has been cited in Introduction of the revised manuscript.*

p.3,l.11: The explanation that the AOM can be eliminated should go into the system description in Sect. 2. A literature review of other methods to sense wind with drones would be very adequate in the introduction. There are a lot more references than Brosy et al. 2017.
*We have revised the manuscript accordingly. The AOM explanation is moved to Section 2, while the Introduction contains a review of different methods of sensing wind with drones and fixed-wing sUAS.*

p.3,l.20: Are measurements above thick forests with a drone lidar with fibre connection to the ground really realistic?
*Due to the indicating location in the manuscript for this comment it seems that the reviewer did not review the second version of our manuscript since the indicated statement appears in the second version of manuscript at p.3, l27. Nevertheless, it is realistic to do such measurements with non-tethered drone which is now stated in the indicated sentence.*

**2.3 Section 2**

p.4-7, Fig. 1-3: Instead of pictures of the single components, it would be good to have a picture with the lidars mounted on the drone gimbal.

*FIgure 2 and Figure 3 show two telescopes mounted on the drone gimbal.*

p.3, l.8: What does 100% duty cycle mean here?

*It means that the laser beams are simultaneously lased through the two telescopes, thus that measurements are performed on two channels continuously and simultaneously (which is stated in the manuscript). As it is not bringinging any additional information '100% duty cycle' has been removed from the manuscript. We are confirming that the reviewer reviewed the first version of the manuscript due to the indicated page and line number of the comment.*

p.5, Tab. 1: Please add lidar wavelength and weight of the telescopes, as well as the focal range that can be adjusted.

*We have added requested information to the revised manuscript.*

p.5, l.9: "... with payload." How big is the payload?

*The stated flight time is with a payload of 5 kg. This information is now enclosed in the text of the manuscript.*

p.5, ll.10ff: If I understand correctly the system has three GNSS-systems, plus another DGNSS system. This seems quite excessive. Isn't the RTK-DGNSS-system more than enough? Real-time DGNSS is not a synonym for RTK. RTK needs a feed of correction data from a nearby base station (not on the rover) to achieve the cm-accuracy. Is this really given in the setup? I think the most important feature are the two antennas for improved pitch and yaw estimation. With the short baseline of only 25 cm it is however quite questionable if this information really improves the IMU-information. Has this been evaluated? Please be more specific in this explanation and distinguish between a dual-antenna setup with a baseline between the two antennas on the rover and an RTK setup with a baseline between base and rover that can be up to a few kilometers and I am not sure if this is really given here. If not (as suggested in the discussion), it is not relevant in this study and should not be "advertised" here.

*The system by default comes with three GNSS-system antennas. The additional two are antennas are part of so-called D-RTK system (differential GPS measurements + RTK) which beside the two additional antennas contains local mobile ground based RTK station (i.e., rover). In our tests we only used differential measurements and not RTK due to some issues in setting up the local rover. Also, the stated accuracy and other drone or gimbal specifications are provided in DJI product sheets which are now refered to.*

p.7, Fig.3: Please indicate the lidar telescopes. Since they look much different than in Fig.1, this is quite confusing. I suggest to remove Fig. 1 and label Fig. 3 accordingly.

*Figure 1 was removed, the telescopes are highlighted on Fig3 (now Fig 2).*

Section 2: How big is the actual take-off weight of the system in this configuration?

*About 3 kg (gimbal + gimbal payload(telescopes, GoPRo camera) + fiber cables )*

**2.4 Section 3**

p.8, ll.1-14: I think it does not really add much to the study to present the preliminary tests with yet another system that has not been fully described, so I suggest to just skip this test and only present the results of the POC system introduced in Sect.2. Figure 4 could be removed in my opinion as well because there is no reference instrument or further evaluation.

*We agree with the reviewer and accordingly we have removed the part of the manuscript related to the preliminary tests.*

p.8, l.22: Does 'VT' and 'TW' have any specific meaning?

*VT stands for V52 Tower (VT), while TW stands for True Wind (TW).*

p.9, l.1: Please mention what kind of sonics are mounted.

*Metek USA-1 3D sonic anemometers*

p.9, l.2f: I am not sure if I understand the positioning procedure. What does it mean that adjustments were done using the GoPro camera?

*DJI Drone has built in camera. However the telescopes were attached to the gimbal which can 'freely' move with respect to the drone. Therefore, we used the drone camera to roughly know that we are close to the reference instrument, while the GoPro camera (which was attached at the tray which carried the telescopes) was used to make sure that we are at the same height as the reference instrument and that we are pointing the laser beams in a right direction (avoiding the beams hitting the mast). We enclosed this clarification in the revised manuscript.*

p.10: I would suggest to make a list of flights that are analyze in this study, with flight time, flight height, telescope orientation and wind and weather conditions.

*New table has been added to the manuscript.*

p.11, l.12: Giving a probed area of 7.07 m x 0.15 m seems a bit awkward and confusing. Maybe it would be easier to understand if you say that homogeneity at the two measured lidar focus points needs to be assumed.

*We agree with the reviewer.*

p.13, ll.11ff: I think this conclusion can not be drawn. The higher $R^2$-value is probably due to the higher number of samples at 70 m. The standard deviation is mostly a measure for turbulence in this case, and in fact the slope of the linear regression is furthest from one. I doubt if a statistical analysis of these very short measurement periods makes any sense at all. I would suggest to show the time series and a comparison of the vertical profiles of tower and drone lidar, including the standard deviation as error bars and quantifying the mean difference instead of the linear regression and point clouds.

*We reformulate the statement regarding why the best comparison is at 70 m according to the reviewer comment. Regarding the presentation of the results in plots we prefer to keep our approach. Besides the linear regression plots we do indeed show the mean difference between the sonic- and drone- based measurements (see Table 4 in the reviewed manuscript).*

p.15, Figs. 11-15: I would suggest to combine these plots in some way. It is probably not necessary to show the single telescope radial wind speeds. I do also not think that the point clouds and linear regressions are necessary and even statistically significant. So, maybe just show the comparison of time series of drone lidar horizontal wind compared to sonic anemometers on all heights in one plot with subplots.

*The indicated figures are combined as a single plot and radial velocities from individual telescopes are removed from plots. We prefer keeping the linear regression plots since they are more indicative on the quantitative aspect of the measurement accuracy, while the time series plots are more useful for qualitative analysis.*

**2.5 Section 4**

p.20, l.11f: I actually disagree and think this is a very good way to show the influence of the out-of-line wind component on the reconstruction from a single beam. I do not think this could be done any better with drone-internal data which is subject to many other uncertainties. However, a comparison of these drone-internal methods with the lidar would be very interesting in future, of course.

*We improved that part of manuscript stating that various methods should be tested to find a suitable one for single beam applications.*

p.20, ll.16ff: Has any indication been found that the downwash distance depends on wind direction and wind speed? If not, this could be a field of further research and areas on to be more conservative in future tests as well.

*We did not investigate the downwash behaviour under different wind conditions. We agree that the future studies should include a dedicated research on the behaviour of the downwash under different conditions (wind speed, wind direction, payload, etc.). We have highlighted this in the revised manuscript.*

p.21, ll.23ff: I would recommend to not describe technical details of possible future developments in a scientific paper. Depending on how developments go, concepts might have to be adapted and it is not really relevant for the scientific ideas and visions

*The reviewer refers to the first version of the manuscript which undergone a brief assessment whether it is suitable for the AMT or not. After that first version of the manuscript we have submitted the second version, which actually does not contain the technical details on the future development.*

**3 Technical corrections**

p.2,l.15: Use the proper text euro sign instead of EURO.

*Similar like the previous comment. i.e. this was already changed in the second version of our manuscript. Therefore, the revised manuscript (version three) also has the correct sign.*

p.3,l.23: The abbreviation POC is only introduced in the abstract, but should be introduced in the main text as well.

*We followed the reviewer recommendation.*

p.4,l.5: There is something wrong here: what is a "demonstrator for an nairborne windspeed requirement"?

*We have removed the indicated part of the sentence.*

p.20,l.9: Some grammar issue here.

*The manuscript was revised accordingly.*

Anonymous Referee #3

*Dear referee,*

*We would like to thank you for the positive review of our paper.*
*We have incorporated your suggestions and made changes to our manuscript.*

*We provide point-by-point responses (in blue) together with the attached revised manuscript which contains highlighted differences between the original and revised manuscript (latexdiff was used for this task).*

p 2-3: the list of criteria, should, at least for EPAs, also include a long duration (months) of the measurements, as a requirement

*We agree with the reviewer and we do indeed discuss the current probl. The manuscript has been updated accordingly.*

Table 1: weight of telescopes?

*The table has been updated with the requested information.*

Fig 7b: I don't understand why they have been put directly in front of each other? Potential flow distortion?

*This is correct, the used setup indeed can induce flow distortion. However, we did not have other options since we used the on-board camera to be sure that the reference sensor is in our viewfield. In future comparison studies we will avoid such setup and use positioning system of the drone itself.*

How to measure profiles?

*With a single telescope: perform sort of a spiral flights with the drone (rotate drone or gimbal 360˚ while moving drone up - down between the desired range of heights). With a dual-telescope: flight the drone up-down between the desired range of heights.*

Other ways of using a drone to measure the wind speed (eg differential motor power).

*This has been now discussed in the introduction.*

[revised manuscript text omitted]

**Review of Robert Sica feedbacks**

This document contains feedbacks from the editor, which are taken as 'screenshots' from the document:
`amt-2019-102-comments-to-author-4.pdf`

The above document contains total of 34 pages, and we will use a page number to emphasize the location of the editor's feedback. For each feedback we provide details how they were addressed in the manuscript revision.

**Pg 1.** There is a red curved line connecting anonymous referee 2 feedback together with our response in blue, as the screenshot of that part of the document indicates:

> Drone-mounted lidar measurements are introduced in this manuscript as a possible replacement for ground-based lidar measurements or meteorological masts. I am very skeptical if this is a realistic scenario, given the logistical constraints that drone measurements have. With this I mean flight time, regulations, operation in all weather conditions (what are maximum wind speeds for example? What about rain?). The authors mention in the conclusion that a lot more development needs to happen before drones can be a serious alternative for power and load verification. However, I think that even with the present research version of this system, a lot of important measurements towards validation of ground-based lidars, studying of wake dynamics and especially turbulence research can be done. I think these applications should be emphasized much more. For this purpose, I wish to see a comparison of turbulence spectra between sonic anemometers and drone lidars for the presented measurements.
>
> *Currently, the drone-based wind lidars cannot replace met masts. However, considering the ongoing developments in drone technology (which is not driven by wind energy or science but due to the need of for example cheaper way of delivering goods) there is potential that this becomes realistic scenario.*
> *The turbulence analysis, indeed of high importance, is not the scope of this paper primarily since the described measurement setup is not suitable to acquired necessary data (e.g., missing accurate information of the gimbal orientation). Nevertheless, it will be investigated in future.*

*We are not sure what the curve line means, thus we did not perform any corresponding changes to our manuscript. A further explanation from the editor would be appreciated in order to improve the manuscript.*

**Pg 7.** The editor requests several changes of the abstract (style + language), as the following screenshot indicates:

> **Abstract.** The fusion of drone and wind lidar technology introduces the exciting possibility of performing high-quality wind measurements virtually anywhere[..[1]]. In this paper we will present a proof of concept (POC) drone-lidar system and report results from several test campaigns that demonstrate its ability to measure accurate wind speeds.
> The POC system is based on a dual-telescope Continuous Wave (CW) lidar, with drone-borne telescopes and ground-based
> 5   opto-electronics. Commercially available drone and gimbal units are employed.
> The demonstration campaigns started with a series of comparisons of the wind speed measurements acquired by the POC system to simultaneous measurements performed by nearby mast based sensors. [..[2]]On average, an agreement to about 0.1 m/s between mast- and drone- based measurements of the horizontal wind speed was found. Subsequently the extent of the flow disturbance caused by the drone downwash was investigated. These tests vindicated the somewhat conservative choice
> 10   of lidar measurement range made for the initial wind speed comparisons.
> Overall, the excellent results obtained without any drone motion correction and with fairly primitive drone position control indicate the potential of drone-lidar systems in terms of accuracy and applications. The next steps in the development are outlined in the paper and several potential applications are discussed.

We have interpreted the editor annotations as the following:
- The abstract should be composed of a single paragraph (i.e., connect paragraphs)
- Sentence 'On average, an agreement to about 0.1 m/s between mast- and drone-based measurements of the horizontal wind speed **was** found' , should be rewritten to 'On average, an agreement to about 0.1 m/s between mast- and drone- based measurements of the horizontal wind speed **is** found'
- Remove 'in the paper' from the last sentence of the abstract

*However, we were not sure why '0.1 m/s' was highlighted. A further explanation from the editor would be appreciated in order to improve the manuscript.*

This is the screenshot of the improved abstract:

**Abstract.** The fusion of drone and wind lidar technology introduces the exciting possibility of performing high-quality wind measurements virtually anywhere. We present a proof of concept (POC) drone-lidar system and report results from several test campaigns that demonstrate its ability to measure accurate wind speeds. The POC system is based on a dual-telescope Continuous Wave (CW) lidar, with drone-borne telescopes and ground-based opto-electronics. Commercially available drone and gimbal units are employed. The demonstration campaigns started with a series of comparisons of the wind speed measurements acquired by the POC system to simultaneous measurements performed by nearby mast based sensors. On average, an agreement to about 0.1 m/s between mast- and drone- based measurements of the horizontal wind speed is found. Subsequently the extent of the flow disturbance caused by the drone downwash was investigated. These tests vindicated the somewhat conservative choice of lidar measurement range made for the initial wind speed comparisons. Overall, the excellent results obtained without any drone motion correction and with fairly primitive drone position control indicate the potential of drone-lidar systems in terms of accuracy and applications. The next steps in the development are outlined and several potential applications are discussed.

**Pg 8.** The editor has requested more appropriated references for atmospheric experiments which were using scanning lidars, as well a reference regarding the impact of the precipitation on the vertical wind speed measured by wind lidars:

[Figure]

The following references were added:
- Regarding the atmospheric experiment which were using lidars:
  McCarthy et al., 1982; Newsom et al., 2008; Collier et al., 2005; Grubišic et al., 2008; Fernando et al., 2015

- Regarding the impact of the precipitation on the vertical wind speed measurements: Lindelöw, 2009

Accordingly, we provided the screenshot of the improved part of the manuscript:

especially to provide wind measurements to properly estimate wind loading on bridges (Cheynet et al., 2017). Historically,
20   (since the 1980s) wind lidars have been used extensively in variety of atmospheric science studies (e.g., McCarthy et al., 1982; Newsom et al., 2008; Collier et al., 2005; Grubišić et al., 2008; Fernando et al., 2015).

Even though wind lidars are cost-attractive instruments for measurements beyond 100-m they are still relatively expensive. A minimum cost of an accurate wind lidar is about 60 k€ and 1-2 M€ for on-shore and offshore applications respectively.

Also, there are circumstances where wind lidars can experience difficulties, in which traditional in-situ measurements can
25   measure successfully. Lidar range is influenced by the atmospheric conditions (i.e., aerosols concentration), which impacts the data availability (in certain locations lidar data availability can fall below mast-based sensor data availability). For example, clouds, fog and snow are highly attenuating for the laser beam which limits the lidar range, and thus data availability. Furthermore, any precipitation will affect the wind speed measurements by lidars. Specifically the vertical component of the wind will be biased since the lidar will dominantly measure the fall velocity of the precipitation (Lindelöw, 2009).

**Pg 9** There is a request for several changes of the introduction text, as indicated by the screenshot below:

– high availability of data (i.e., not to be hindered by fog, low clouds, etc.).

– long measurement duration (e.g., months)

Upper Case S

A potential solution for the above formulated problem is to use a [..⁵ ]Small Unmanned Aircraft System (sUAS) (such as
20   multi-copter drones) as a platform for a wind lidar [..⁶ ]even though currently sUAS cannot offer a long uninterrupted operation. Typically sUAS acquire wind speed information either by utilizing flow sensors such multi-hole pitot tube probes (e.g., Wildmann et al., 2014) or sonic anemometers (e.g., Nolan et al., 2018) or without flow sensors by measurements and conversion of aircraft dynamics (e.g. Neumann and Bartholmai, 2015). For example, studies Neumann and Bartholmai (2015), Palomaki et al. (2017) and Brosy et al. (2017) utilized real-time measurements of multi-copter dynamics to estimate
25   wind speed. These studies reported a good agreement of the estimated wind speed with the speed measured by mast-mounted sonic anemometers, where the sonic and estimated wind speed agreed to about 0.5 to 0.7 m/s for 10 s averaged data (Palomaki et al., 2017; Brosy et al., 2017) and 0.3 m/s for 20 s averaged data (Neumann and Bartholmai, 2015). Brosy et al. (2017) stated that the wind speed estimated using only drone dynamics should not be used as information about atmospheric turbulence since due to the volume drones don't react to the small eddies, and thus this approach
30   cannot capture a full range of wind speeds. In LAPSE-RATE experiment (Barbieri et al., 2019) several multi-copter drones were equipped with sonic anemometers. The calibration flights of such drones (which entail hovering the equipped

⁵removed: drone as
⁶removed: . This idea

We have performed the following changes in our manuscript:
- Changed sUAS to SUAS
- Removed brackets around text 'such as multi-copter drones'
- Did not removed 'even though'
- We were not sure what 'c' means
- It was not clear how to interpret 'by', should it be placed after 'studies' ?

The screenshot of the changed manuscript is provided below:

> 15    A potential solution for the above formulated problem is to use a Small Unmanned Aircraft System (SUAS), such as multi-copter drones, as a platform for a wind lidar even though currently SUAS cannot offer a long uninterrupted operation. Typically SUAS acquire wind speed information either by utilizing flow sensors such multi-hole pitot tube probes (e.g., Wildmann et al., 2014) or sonic anemometers (e.g., Nolan et al., 2018) or without flow sensors by measurements and conversion of aircraft dynamics (e.g. Neumann and Bartholmai, 2015). For example, studies Neumann and Bartholmai (2015), Palomaki
>
> 20    et al. (2017) and Brosy et al. (2017) utilized real-time measurements of multi-copter dynamics to estimate wind speed. These studies reported a good agreement of the estimated wind speed with the speed measured by mast-mounted sonic anemometers, where the sonic and estimated wind speed agreed to about 0.5 to 0.7 m/s for 10 s averaged data (Palomaki et al., 2017; Brosy et al., 2017) and 0.3 m/s for 20 s averaged data (Neumann and Bartholmai, 2015). Brosy et al. (2017) stated that the wind speed

**Pg 10.** The sentence 'However, if we use a wind lidar as a flow sensor, potentially the accuracy could be improved and also ability to acquire turbulence measurements' was requested to be changed, by inserting 'given the' in front of 'ability to acquire turbulence measurements':

> drones close to masts equipped with sonic anemometers) showed agreement of the mast and drone-based 15-s averaged wind speed measurements to about 0.75 m/s (Nolan et al., 2018). Like in case of multi-copter drones, the aircraft dynamics of fixed-wing sUAS can be used to determine wind speed (Rautenberg et al., 2018) with the wind speed accuracy generally worse than multi-copter drones (Barbieri et al., 2019). As stated in Rautenberg et al. (2018) utilizing a
>
> 5    flow sensor such as pitot tube on-board of fixed-wing sUAS generally provides better results. Nevertheless, as reported in Barbieri et al. (2019) an average accuracy of sUAS of both fixed-wing and multi-copter concepts with or without flow sensors is about 0.5 m/s, which for the majority of wind energy applications is not sufficient.
>
>     However, if we use a wind lidar as a flow sensor, potentially the accuracy could be improved and also ability to acquire turbulence measurements. Equiping sUAS with wind lidars has been suggested in an early study of using [..[7]]sUAS for wind
>
> 10    energy applications (Giebel et al., 2012). At the time of the study, it was technically unfeasible to pursue this idea. Roughly a decade later, both the lidar and drone technology have advanced significantly unlocking the potential to explore the proposed idea.

We did introduce this change, however, potentially we've misinterpreted the editor feedback. In our opinion by introducing this modification the revised sentence does not read well. A further explanation from the editor would be appreciated in order to improve the manuscript.

**Pg 16.** The editor indicates problems with formatting, as the screenshot below indicates:

> experiments; the second channel was left unconnected. The measurements were carried out close to Malvern U.K. in July 2018, and provided important experience that was carried forward to the trials in Denmark. The biggest challenge involved handling of the 100-m fibres to avoid snagging, which could lead to bend loss and disturbance of the drone 's flight. Careful stacking of the uncoiled fibres on the ground was essential, with separation of the T (transmitting) and R (receiving) fibres to minimise risk of tangling.
>
> [31]removed: Wind measurements were made by manually rotating the drone to point the telescope into the wind. The weather was hot and sunny with light winds (5 m/s and lower) from the south-southwest (SSW) direction. High quality 50 Hz LOS wind data were obtained over the full flight duration at heights of up to 70 m above ground level. Figure **??** shows a time series covering the middle seven minutes of the flight, when the beam was clear of any obstructions such as trees and buildings. The experiment provided encouragement for the viability of drone-mounted wind lidar, and notably there were no observed problems caused by vibrations of the lidar telescope on the drone . No obvious showstoppers were identified from this preliminary demonstration, and this provided the impetus to plan the next set of more rigorous tests in Denmark.
>
> [32]removed: Time series showing measured LOS wind speed from preliminary drone-mounted lidar experiment
>
> [33]removed: As earlier mentioned, the main campaigns, which will be described in the following subsections of this paper, were performed with Matrice 600 Pro, thus with the system described in Section 2 of the paper.
>
> [34]removed: The fine adjustment of the drone position and gimbal orientation were done using the GoPro camera mounted on the tray
>
> FORMATING?
>
> 10

As mentioned in our email correspondence this is mainly the product of latexdiff command which was positioning a part of the manuscript that was removed from the original text in the footer of the compiled pdf. The latest latexdiff command does not do that, but instead it

keeps the removed text while using  to indicate that it has been removed in the revised manuscript.

**Pg 29** The editor requests changes of the sentence 'The generation of accurate turbulence data from drone-mounted lidar requires knowledge of the telescope's orientation and speed to allow motion compensation, and this will form the topic for a future paper', which can be seen in the screenshot below:

The editor request was understood as following:
- remove 'a' in front of 'future paper'.
- replace 'paper' by 'study'

Based on this interpretation, the manuscript was improved as it can be seen from the screenshot below: